

**Weakened aerosol-radiation interaction exacerbating ozone**
**pollution in eastern China since China's clean air actions**
Hao Yang[1,2], Lei Chen[1], Hong Liao[1], Jia Zhu[1], Wenjie Wang[3], Xin Li[3]
[1]Jiangsu Key Laboratory of Atmospheric Environment Monitoring and Pollution
Control, Jiangsu Collaborative Innovation Center of Atmospheric Environment and
Equipment Technology, School of Environmental Science and Engineering, Nanjing
University of Information Science & Technology, Nanjing 210044, China
[2]College of Materials Science and Engineering, Guizhou Minzu University, Guiyang
550025, China
[3]State Joint Key Laboratory of Environmental Simulation and Pollution Control,
College of Environmental Sciences and Engineering, Peking University, Beijing
100871, China
**Correspondence to:** Lei Chen (chenlei@nuist.edu.cn) and Hong Liao
(hongliao@nuist.edu.cn)



## Abstract

Since China's clean air action, $PM_{2.5}$ air quality has been improved while ozone ($O_3$) pollution has been becoming severe. Here we apply a coupled meteorology-chemistry model (WRF-Chem) to quantify the responses of aerosol-radiation interaction (ARI), including aerosol-photolysis interaction (API) and aerosol-radiation feedback (ARF), to anthropogenic emission reductions from 2013 to 2017, and their contributions to $O_3$ increases over eastern China in summer and winter. Sensitivity experiments show that the decreased anthropogenic emissions play a more prominent role for the increased MDA8 $O_3$ both in summer (+1.96 ppb vs. +0.07 ppb) and winter (+3.56 ppb vs. -1.08 ppb) than the impacts of changed meteorological conditions. The decreased $PM_{2.5}$ caused by emission reduction can result in a weaker impact of ARI on $O_3$ concentrations, which poses a superimposed effect on the worsened $O_3$ air quality. The weakened ARI due to decreased anthropogenic emission aggravates the summer (winter) $O_3$ pollution by +0.81 ppb (+0.63 ppb) averaged over eastern China, with weakened API and ARF contributing 55.6% (61.9%) and 44.4% (38.1%), respectively; this superimposed effect is more significant for urban areas during summer (+1.77 ppb). Process analysis indicates that the enhanced chemical production is the dominant process for the increased $O_3$ concentrations caused by weakened ARI both in summer and winter. This study innovatively reveals the adverse effect of weakened aerosol-radiation interaction due to decreased anthropogenic emissions on $O_3$ air quality; more stringent coordinated air pollution control strategies are needed for future air quality improvement.





## 1. Introduction

With the implementation of clean air action since 2013, PM$_{2.5}$ (particulate matter with an aerodynamic equivalent diameter of 2.5 micrometers or less) concentrations have decreased significantly in China (Zhai et al., 2019; Zhang et al., 2019). However, ozone (O$_3$) pollution is becoming worse and poses a significant challenge over eastern China, especially in the developed city clusters including Beijing-Tianjin-Hebei (BTH), Yangtze River Delta (YRD), Pearl River Delta (PRD), and Sichuan Basin (SCB) (Lu et al., 2018; Dang and Liao, 2019; Li et al., 2019; Li et al., 2021). According to observation data, Li et al. (2020) found that the daily maximum 8-h average O$_3$ concentrations (MDA8 O$_3$) increased at a rate of 1.9 ppb a$^{-1}$ from 2013 to 2019 over eastern China. Elevated O$_3$ concentrations can not only decrease crop yield but also damage human health (Lelieveld et al., 2015; Yue et al., 2017; Mills et al., 2018). Therefore, it is essential to gain a comprehensive understanding about factors driving the increasing trend of O$_3$ in China in order to formulate effective prevention strategies.

As a secondary air pollutant, troposphere O$_3$ can be produced by nitrogen oxides (NO$_x$ = NO + NO$_2$) and volatile organic compounds (VOCs) in the presence of solar radiation through photochemical reactions (Atkinson, 2000; Seinfeld and Pandis, 2006). Consequently, the concentration of O$_3$ is closely related to changes in meteorological conditions and anthropogenic emissions (Wang et al., 2019; Liu and Wang, 2020a,b; Shu et al., 2020). Moreover, particulates can also affect O$_3$ concentrations through aerosol-radiation interaction (ARI), including aerosol-photolysis interaction (API) and aerosol-radiation feedback (ARF) (Liao et al., 1999; Wang et al., 2016; Zhu et al., 2021; Yang et al., 2022), and heterogeneous chemistry on aerosol surface (Lou et al., 2014; Li et al., 2019; Liu and Wang, 2020b). Many studies have found that the decreased PM$_{2.5}$ can be one of the driving factors contributing to the increased O$_3$ concentrations (Li et al., 2019; Liu and Wang, 2020b; Shao et al., 2021). Li et al. (2019) analyzed GEOS-Chem simulation results and pointed out that the reductions in PM$_{2.5}$ concentrations from 2013 to 2017 in North China Plain (NCP) could decrease the sink of HO$_2$ on aerosol surface, which would result in the increase in O$_3$ concentrations.



When heterogeneous reactions were considered in WRF-CMAQ, Liu and Wang (2020b)
found that decreased $PM_{2.5}$ concentrations weakened the uptake of reactive gases
(mainly $HO_2$ and $O_3$) which led to the increase in $O_3$ concentrations over China from
2013 to 2017. However, the contribution of weakened aerosol-radiation interaction due
to substantial decreases in $PM_{2.5}$ under clean air action to the increased $O_3$ has not been
systematically quantified. Furthermore, previous studies mainly focus on the increased
summer $O_3$ (Li et al., 2019; Liu and Wang, 2020a,b; Shu et al., 2020; Shao et al., 2021),
but underlying reasons driven the changes in winter $O_3$ is unclear. Li et al. (2021)
pointed out that $O_3$ pollution has been extended into cold seasons under the emission
control measures. Therefore, this study aims to quantify the response of aerosol-
radiation interaction to anthropogenic emission reduction from 2013 to 2017, with the
mainly focus on the contribution to changed $O_3$ concentrations over eastern China both
in summer and winter.

Aerosol-radiation interaction (ARI) can alter photolysis rates through aerosol-

photolysis interaction (API) and meteorological variables through aerosol-radiation
feedback (ARF) to suppress $O_3$ formation (Yang et al., 2022). Hong et al. (2020) used
WRF-CMAQ in conjunction with future emission scenarios to find that weakened ARF
due to reduced aerosol concentration led to an increase in the daily maximum 1-h
average $O_3$ concentration in eastern China from 2010 to 2050. By using WRF-CMAQ,
Liu and Wang (2020b) reported that weakened API could increase the MDA8 $O_3$
concentrations by 0.3 ppb in urban areas from 2013 to 2017. Zhu et al. (2021) used
WRF-Chem to investigate the impact of weakened ARF on air pollutants over NCP
during COVID-19 lockdown and reported that the weakened ARF would increase the
$O_3$ concentrations by 7.8%. In general, previous studies mainly examined the impact of
either weakened ARF or API, systematic analysis about the total and the respective
impacts of changed API and/or ARF on $O_3$ over eastern China both in summer and
winter from 2013 to 2017 have not been conducted.

The objective of this manuscript is to examine the impacts of aerosol-radiation

interactions (ARI), including the effects of aerosol-photolysis interaction (API) and
aerosol-radiation feedback (ARF), on $O_3$ concentrations over eastern China both in



summer and winter by using the online coupled WRF-Chem model, with the main focus
on their responses to clean air action. Process analysis is also applied to explore the
prominent physical/chemical process responsible for the changed impacts of API and/or
ARF on surface $O_3$. This study is believed to provide insights into the role of weakened
ARI on $O_3$ levels over eastern China not only in summer, but also in winter. In Section
2, we describe the model configuration, numerical experiments, observational data, and
the integrated process rate analysis. Model evaluation is presented in Section 3. The
presentation of model results and the corresponding analyses are exhibited in Section
4. Conclusions are provided in Section 5.

## 2. Methodology

2.1 Model configuration

The model used in this study is an online-coupled meteorology-chemistry model,

Weather Research and Forecasting with Chemistry model (WRF-Chem v3.7.1), that
can simulate meteorological fields and concentrations of gases and aerosols
simultaneously (Grell et al., 2005; Skamarock et al., 2008). Figure S1 shows the
simulated domain that covers most regions of China with a horizontal resolution of 27
km and grid points of 167 (west–east) × 167 (south–north). The model contains 32
vertical levels extending from the surface to 50 hPa, with the first 16 layers located
below 2 km to resolve fine boundary layer processes. The enclosed black line in Figure
S1 represents the eastern China (22-41.5 °N, 102-123 °E), and the four heavily polluted
regions are also selected for analysis, including BTH (36.0-41.5 °N, 113-119.5 °E),
YRD (29.5-32.5 °N, 118-122 °E), PRD (21-23.5 °N, 112-116 °E), and SCB (27.5-
31.5 °N, 102.5-107.5 °E), respectively.

The National Center for Environmental Prediction (NCEP) Final Analysis dataset

(FNL) with a spatial resolution of 1° × 1° and 6-hour temporal resolution are used to
provide the meteorological initial and lateral boundary conditions. The chemical initial
and boundary conditions for the WRF-Chem model are taken from the outputs of
Community Atmosphere Model with Chemistry (CAM-Chem).

The Carbon Bond Mechanism Z (CBM-Z) is applied as the gas-phase chemical





mechanism (Zaveri and Peters, 1999), and the full 8-bin MOSAIC (Model for
Simulating Aerosol Interactions and Chemistry) aerosol module with aqueous
chemistry is used to simulate aerosol evolution (Zaveri et al., 2008). In MOSAIC
module, aerosols are assumed to be internally mixed into 8 bins (0.039–0.078 μm,
0.078–0.156 μm, 0.156–0.312 μm, 0.312–0.625 μm, 0.625–1.25 μm, 1.25–2.5 μm, 2.5–
5.0 μm and 5.0–10 μm), and each bin considers all major aerosol species, such as sulfate
($SO_4^{2-}$), nitrate ($NO_3^-$), ammonium ($NH_4^+$), black carbon (BC), organic carbon (OC), and
other inorganic mass (secondary organic aerosols are not included in MOSAIC (Yang
et al., 2022)). The impacts of aerosols on photolysis rates are calculated by using the
Fast-J scheme (Wild et al., 2000). The following physical parameterizations are used in
WRF-Chem. The Rapid Radiative Transfer Model for general circulation models
(RRTMG) scheme is used to treat both shortwave and longwave radiation in the
atmosphere (Iacono et al., 2008). The Purdue Lin microphysics scheme (Lin et al., 1983)
and the Grell 3D ensemble scheme (Grell, 1993) are used to describe the cloud
microphysical and cumulus convective processes. The Noah land surface scheme (Chen
and Dudhia, 2001) and the Monin-Obukhov surface scheme (Foken, 2006) are used to
simulate land-atmosphere interactions. The planetary boundary layer is characterized
by Yonsei University PBL scheme (Hong et al 2006).
In this study, Multi-resolution Emission Inventory for China (MEIC;
http://www.meicmodel.org/) in 2013 and 2017 are used as the anthropogenic emissions
of particles and gases (Zheng et al., 2018). Biogenic emissions are calculated online by
using the Model of Emissions of Gases and Aerosols from Nature (MEGAN) developed
by Guenther et al. (2006).
2.2 Numerical experiments
Seven sensitivity experiments are designed (Table 1). Here are the detailed
descriptions:
(1) BASE_17E17M: This baseline experiment is coupled with the interactions
between aerosol and radiation, which includes the impacts of API and ARF. Both
the meteorological field and anthropogenic emission are fixed at year 2017.



(2) BASE_13E13M: Same as BASE_17E17M, but the meteorological field and anthropogenic emission are fixed at year 2013.

(3) NOAPI_17E17M: Same as BASE_17E17M, but the impact of API is not considered by turning off the aerosol effect in the photolysis module, following the method described in Yang et al. (2022).

(4) NOALL_17E17M: Same as BASE_17E17M, but neither the impact of API nor ARF is considered by zeroing the aerosol optical properties in the optical module, following the method described in Yang et al. (2022).

(5) BASE_13E17M: Same as BASE_17E17M, but the anthropogenic emission is fixed at year 2013.

(6) NOAPI_13E17M: Same as NOAPI_17E17M, but the anthropogenic emission is fixed at year 2013.

(7) NOALL_13E17M: Same as NOALL_17E17M, but the anthropogenic emission is fixed at year 2013.

Figure 1 detailedly presents the schematic overview of designed numerical experiments. As shown in Fig. 1, the differences between BASE_17E17M and BASE_13E13M (BASE_17E17M minus BASE_13E13M) represent the changed $O_3$ ($\Delta O_3$) due to variations in meteorology and anthropogenic emissions from 2013 to 2017. The differences between BASE_13E17M and BASE_13E13M (BASE_13E17M minus BASE_13E13M) show the impact of changed meteorological conditions on $O_3$ ($\Delta O_3\_MET$) from 2013 to 2017. The differences between BASE_17E17M and BASE_13E17M (BASE_17E17M minus BASE_13E17M) indicate the impact of anthropogenic emission reductions on $O_3$ ($\Delta O_3\_EMI$) from 2013 to 2017.

The impacts of aerosol-radiation interaction (ARI) on $O_3$ under different anthropogenic emission scenarios (i.e., strong anthropogenic emission levels in year 2013, and weaker anthropogenic emission levels in year 2017) can be analyzed as the differences between BASE_17E17M and NOALL_17E17M (BASE_17E17M minus NOALL_17E17M, denote as $\Delta O_3\_ARI_{17E}$), and BASE_13E17M and NOALL_13E17M (BASE_13E17M minus NOALL_13E17M, denote as $\Delta O_3\_ARI_{13E}$).




Thus, the impact of weakened ARI due to clean air action on $O_3$ (denote as
$\Delta O_3\_\Delta ARI\_EMI$) can be quantified from the differences between $\Delta O_3\_ARI_{17E}$ and
$\Delta O_3\_ARI_{13E}$. Similarly, the impacts of weakened API (denote as $\Delta O_3\_\Delta API\_EMI$) and
ARF (denote as $\Delta O_3\_\Delta ARF\_EMI$) due to decreased anthropogenic emission on $O_3$ can
also be estimated from the differences between (BASE_17E17M minus
NOAPI_17E17M, denote as $\Delta O_3\_API_{17E}$) and (BASE_13E17M minus
NOAPI_13E17M, denote as $\Delta O_3\_API_{13E}$), and between (NOAPI_17E17M minus
NOALL_17E17M, denote as $\Delta O_3\_ARF_{17E}$) and (NOAPI_13E17M minus
NOALL_13E17M, denote as $\Delta O_3\_ARF_{13E}$), respectively. Detailed descriptions can be
found in Fig. 1.
Simulation periods are integrated from 30 May to 30 June (denoted as summer)
and 29 November to 31 December (denoted as winter) both in 2013 and 2017. To avoid
potential deviations caused by long-term model integration, each simulation is re-
initialized every eight days, with the first 40 hours as the model spin-up. The complete
simulation includes five model cycles. Simulation results from the BASE_17E17M
case during summer and winter are used to evaluate the model performance. If not
otherwise specified, the time in this paper is the local time, and the synergetic impacts
of ARF and API are equal to the impact of ARI (i.e., ARI=ARF+API).
2.3 Observational data
Meteorological observations of temperature ($T_2$), relative humidity ($RH_2$), wind
speed ($WS_{10}$) and wind direction ($WD_{10}$) provided by the NOAA's National Climatic
Data Center (https://www.ncei.noaa.gov/) are used to validate the model
meteorological performance. In this study, 353 stations are selected and the locations
are shown as red dots in Fig. S1. Observed surface $PM_{2.5}$, $O_3$ and $NO_2$ concentrations
in eastern China are obtained from the China National Environmental Monitoring
Center, which can be downloaded from http://beijingair.sinaapp.com. To ensure the
data quality, a single site with at least 500 actual observations during the simulated
period are used for model evaluation. A total of 1296 sites, as shown in Fig. 2a, are
obtained. Photolysis rates of nitrogen dioxide ($NO_2$) ($J[NO_2]$) measured at the Peking



University site (39.99 °N, 116.31 °E) are also used to evaluate the model performance.
2.4 Integrated process rate analysis
In order to quantitatively elucidate individual contributions of physical and
chemical processes to $O_3$ concentration changes due to weakened ARI, the integrated
process rate (IPR) methodology is applied in this study. IPR analysis is an advanced
tool to evaluate the key process for $O_3$ concentration variation (Shu et al., 2016; Zhu et
al., 2021; Yang et al., 2022). In this study, the IPR analysis tracks hourly (e.g., one time
step) contribution to $O_3$ concentration variation from four main processes, including
vertical mixing (VMIX), net chemical production (CHEM), horizontal advection
(ADVH), and vertical advection (ADVZ). We define ADV as the sum of ADVH and
ADVZ.
**3. Model Evaluation**
Simulation results of BASE_17E17M are used to compare with the observations
to evaluate the model performs before interpreting the impacts of aerosol-radiation
interaction on surface-layer ozone concentration.
3.1 Evaluation for meteorology
Figure S2 shows the time series of observed and simulated $T_2$, $RH_2$, $WS_{10}$, and
$WD_{10}$ averaged over the 353 meteorological stations in China during summer and
winter in 2017. Statistical performances of simulated meteorological parameters
compared with ground-based observations are shown in Table 2. Simulations track well
with observed $T_2$ with the correlation coefficient (R) of 0.99 and 0.92, but underestimate
$T_2$ with the mean bias (MB) of -1.0 and -2.0 K in summer and winter, respectively.
Simulated $RH_2$ agree reasonably well with observations with R of 0.97 and 0.87, and
small normalized mean biases (NMB) are found in summer and winter with values of
3.2% and 3.5%, respectively. $WS_{10}$ is slightly overpredicted with the MB of 1.6-2.1 m
$s^{-1}$. The R and root-mean-square error (RMSE) of $WS_{10}$ are 0.77-0.82 and 1.6-2.1 m $s^{-}$
$^1$, respectively. Large bias in wind speed can be partly caused by unresolved
topographical features (Jimenez and Dudhia, 2012). The NMB of $WD_{10}$ ranges from -





3.9% to -2.6% and the R ranges from 0.40 to 0.69, respectively. As shown in Fig. S3,
the predicted J[$NO_2$] match well with the observations with R of 0.93-0.94 and NMB
of 4.8%-12.3%. In general, the simulated meteorological variables fairly well
agreement with the observations.
3.2 Evaluation for air pollutants
Figure 2 shows the spatial-temporal variations of observed and simulated near-
surface $PM_{2.5}$, $O_3$ and $NO_2$ concentrations averaged over eastern China during summer
and winter in 2017. As demonstrated in Figs. 2(a1) and (c1), WRF-Chem model
reasonably well reproduces the spatial distribution of observed $PM_{2.5}$, with high values
over large city cluster. The predicted $O_3$ concentrations can also reproduce the spatial
variation of the observed concentrations (Figs. 2(a2) and (c2)). $NO_2$ is an important
precursor of $O_3$ and aerosol, a good performance on $NO_2$ is necessary. From Figs. 2(a3)
and (c3), the model can well reproduce the spatial distribution of observed $NO_2$.
Although the distributions of simulated air pollutants are in good with the observations,
biases still exist, which may be due to the uncertain in the emission inventories. Figures
2(b1-b3) and 2(d1-d3) show the temporal profiles of observed and simulated surface-
layer air pollutants averaged over monitoring sites and the grid cell containing the
monitor site in eastern China. The statistical metrics are also shown in Table 2. As
shown in Figs. 2(b1) and (d1), the model tracks well with the diurnal variation of $PM_{2.5}$
over the eastern China, with R of 0.63 and 0.80, respectively. But the model slightly
underestimates the concentrations of $PM_{2.5}$ with MB of -6.3 and -10.1 µg m$^{-3}$,
respectively, in summer and winter. Simulated $O_3$ agree reasonably well with
observations with R of 0.90 and 0.86, and small MB are found in summer and winter
with values of -0.6 and 2.8 ppb, respectively. The model tracks the daily variation of
observed $NO_2$ reasonably well, with R of 0.73 and 0.83. But the model slightly
underestimates the $NO_2$ against measurements, with MB of -1.5 and -4.5 ppb,
respectively, in summer and winter. In general, WRF-Chem model can well reproduce
the features of observed meteorology and air pollutants over eastern China.





## 4. Results and Discussion

4.1 Impacts of changed meteorology and anthropogenic emission on $O_3$

The strategy of clean air action decreased the anthropogenic emission of $NO_x$, but the changes in anthropogenic VOCs emissions were unobvious (Fig. S4), which might influence the $O_3$ formation sensitive regime and the $O_3$ concentration. Figure 3 shows the spatial distributions of changed summer and winter MDA8 $O_3$ concentrations from 2013 to 2017 over eastern China, and the contributions of changed anthropogenic emissions alone and changed meteorological conditions alone. As shown in Fig. 3(b), the concentration of summer MDA8 $O_3$ from 2013 to 2017 was increased in city clusters, but it was decreased in rural regions. This discrepancy might be explained by the ozone formation regimes in urban (typically VOCs-limited) and rural (typically $NO_x$-limited) areas during summer (Li et al., 2019; Wang et al., 2019). Contrary to the phenomenon in summer, decreased anthropogenic emissions lead to a uniform increase in winter MDA8 $O_3$ over the whole eastern China (Fig. 3(e)). The different spatial variation characteristics in summer and winter could be explained by the different ozone formation regimes in winter (VOCs-limited) and summer ($NO_x$-limited) (Jin and Holloway, 2015). From Figs. 3(c) and (f), the impacts of changed meteorological conditions on MDA8 $O_3$ varied by regions, ranging from -24.9 (-14.0) to 17.0 (7.3) ppb in summer (winter).

The reductions in anthropogenic emissions from 2013 to 2017 will also lead to a decrease in $PM_{2.5}$ concentrations (Fig. S5), which can further affect the $O_3$ concentrations by weakened aerosol-radiation interaction (ARI). Further, we average the observed MDA8 $O_3$ concentrations of monitoring sites in the urban areas and the simulation value for the grid cell containing the monitoring site to examine the impacts of changed meteorological conditions, anthropogenic emissions and ARI on $O_3$ levels in densely populated urban areas (Fig. 4). Given that most of the monitoring stations with 5 years of continuous observations are located in urban areas. Therefore, these monitoring stations and the grid cells containing the monitoring stations can be considered as urban areas in this study (Liu and Wang, 2020b). As shown in Figs. 4(a1)



and (b1), the changes in observed MDA8 $O_3$ over urban areas in eastern China from
2013 to 2017 can be well captured by WRF-Chem both in summer and winter. In
summer, changed meteorological conditions from 2013 to 2017 has little impact on the
variations in MDA8 $O_3$ over the urban areas, while the contribution of emission
reductions to increased MDA8 $O_3$ is significant. In winter, changed meteorological
conditions is unfavorable for the increase in MDA8 $O_3$ from 2013 to 2017, indicating
the worsened ozone pollution driven by the changed anthropogenic emission. What's
more, the $\Delta O_3\_\Delta ARI\_EMI$ has significant effect on the increased MDA8 $O_3$ in summer
from 2013 to 2017 with the value of +1.77 ppb (87.6%), but its impacts in winter are
smaller, only +0.42 ppb (11.8%), which is consistent with the results in Li et al. (2021).
Meanwhile, the contributions of $\Delta O_3\_\Delta API\_EMI$ and $\Delta O_3\_\Delta ARF\_EMI$ to the increase
in $O_3$ concentration averaged over urban areas in eastern China are almost the same in
summer (0.79 vs. 0.98) and winter (0.20 vs. 0.22). The model can also capture the
changes in observed summer/winter MDA8 $O_3$ from 2013 to 2017 over urban areas in
the four city clusters (Figs. 4(a2-b5)), except BTH in summer. The reason for the
underestimation over BTH may be that this study did not consider the effect of changes
in aerosol heterogeneous reactions. Li et al. (2019) found that the weakened uptake of
$HO_2$ on aerosol surfaces was the main reason for the $O_3$ increase over BTH. In general,
we find that the enhancement of $O_3$ concentrations both in summer and winter is mainly
caused by the factor of reduced anthropogenic emissions. Furthermore, the
contributions of $\Delta O_3\_\Delta API\_EMI$ and $\Delta O_3\_\Delta ARF\_EMI$ to the increases in $O_3$
concentrations from 2013 to 2017 over urban areas are almost the same during summer
and winter.
4.2 Impacts of weakened aerosol-radiation interaction on $O_3$
Figures S6a (S7a) and S6b (S7b) present the spatial distribution of the impacts of
ARF, API and ARI on surface MDA8 $O_3$ concentrations in summer (winter) under
different anthropogenic emission conditions in year 2017 and 2013, respectively. As
shown in Fig. S6, summer MDA8 $O_3$ are significantly reduced over eastern China, ARF,



API and ARI decrease the surface MDA8 $O_3$ concentrations by 0.23 (0.59) ppb, 1.09
(1.54) ppb and 1.32 (2.13) ppb under low (high) anthropogenic emission conditions in
year 2017 (year 2013), respectively. The changes in MDA8 $O_3$ concentrations due to
aerosol-radiation interaction under low emission condition are weaker than that under
high emission condition. This is because the concentration of aerosols in year 2013 is
higher than that in year 2017, and then its impact on meteorological conditions and
$J[NO_2]$ is greater (Fig. S8). As shown in Fig. S7a, ARF, API and ARI decrease the
winter MDA8 $O_3$ concentrations by 0.38 ppb (-0.9%), 1.59 ppb (-4.1%) and 1.96 ppb
(-5.1%) in year 2017, respectively. Compared to the impacts under relatively high
anthropogenic emission conditions in year 2013, the reduction of surface MDA8 $O_3$
concentrations caused by ARF, API and ARI are also greater, with the values of 0.62
ppb (-1.6%), 1.98 ppb (-5.4%) and 2.59 ppb (-7.1%), respectively. Both API and ARF
reduce $O_3$ concentrations, and the reduction in $O_3$ caused by API is greater than that
caused by ARF both in summer and winter.

Further, the significant reduction in $PM_{2.5}$ due to clean air action (Fig. S5) will

lead to an increase in $O_3$ concentrations as the weakened effects of aerosols on $O_3$.
Therefore, this study further quantifies the effects of $\Delta O_3\_\Delta API\_EMI$,
$\Delta O_3\_\Delta ARF\_EMI$ and $\Delta O_3\_\Delta ARI\_EMI$ on $O_3$ air quality. As shown in Figs. 5(a1-a3),
the surface MDA8 $O_3$ in summer are increased over most of eastern China due to
$\Delta O_3\_\Delta API\_EMI$, $\Delta O_3\_\Delta ARF\_EMI$ and $\Delta O_3\_\Delta ARI\_EMI$. The largest increases in
MDA8 $O_3$ concentrations due to $\Delta O_3\_\Delta API\_EMI$ and $\Delta O_3\_\Delta ARF\_EMI$ are found in
the developed four city clusters, with the increase larger than 4 ppb. Overall,
$\Delta O_3\_\Delta API\_EMI$, $\Delta O_3\_\Delta ARF\_EMI$ and $\Delta O_3\_\Delta ARI\_EMI$ lead to the increase in
surface MDA8 $O_3$ by 0.36 ppb, 0.45 ppb and 0.81 ppb averaged over eastern China
during summer, respectively. As shown in Fig. 5(a4-a6), the $\Delta O_3\_\Delta API\_EMI$,
$\Delta O_3\_\Delta ARF\_EMI$ and $\Delta O_3\_\Delta ARI\_EMI$ can also cause an increase in winter MDA8 $O_3$
concentrations by 0.24 ppb, 0.39 ppb and 0.63 ppb, respectively. In general, weakened
aerosol-radiation interaction due to reduced anthropogenic emission from 2013 to 2017
can exacerbate ozone pollution both in summer and winter.

In order to explore the mechanism of the impacts of $\Delta O_3\_\Delta ARI\_EMI$ on MDA8




$O_3$, we resolve the changed $O_3$ into the contributions from chemical and physical
processes. Figure 6 presents the accumulated changes in $O_3$ and each process
contribution from 09:00 to 16:00 LST by the $\Delta O_3\_\Delta API\_EMI$, $\Delta O_3\_\Delta ARF\_EMI$ and
$\Delta O_3\_\Delta ARI\_EMI$    ($\Delta O_3\_\Delta ARI\_EMI = \Delta O_3\_\Delta API\_EMI + \Delta O_3\_\Delta ARF\_EMI$)    during
summer and winter. As shown in Fig 6, the enhanced chemical production is the
dominant process leading to the increase in $O_3$ concentrations over eastern China and
the four city clusters both in summer and winter. The leading factor of enhancement in
$O_3$ over BTH are inconsistent with that over eastern China, and the enhancement of $O_3$
concentration in BTH is mainly due to $\Delta O_3\_\Delta ARF\_EMI$. But the leading factor of
enhancement in $O_3$ over SCB are consistent with that in eastern China, the enhancement
of $O_3$ concentration is mainly due to $\Delta O_3\_\Delta API\_EMI$ both in summer and winter.
Moreover, the enhancement of $O_3$ concentration in BTH, YRD and PRD is mainly due
to $\Delta O_3\_\Delta ARF\_EMI$ during winter, which is opposite to that of eastern China. The
leading factors for the increase of $O_3$ concentration in different city clusters are different.
The enhancement of $O_3$ concentration in most areas is caused by $\Delta O_3\_\Delta API\_EMI$,
whereas the increase in $O_3$ concentration in BTH, YRD and PRD areas is dominated by
$\Delta O_3\_\Delta ARF\_EMI$ in winter. In general, the weakened aerosol-radiation interaction
caused by emission reduction would promote the chemical production of $O_3$ and
increase the $O_3$ concentrations over eastern China in summer and winter.
In order to explore the reason for the increase in $O_3$ chemical production, we
further analyzed the variation of $HO_x$ ($HO + HO_2$) concentration from 2013 to 2017. As
the aerosol concentration decreases, its influence on solar radiation is weakened and
photolysis is enhanced, leading to an increase in $HO_x$ levels. It can be seen from Fig.
S9 that the concentration of $HO_x$ increases both in winter and summer. The increase in
$HO_x$ will promote the conversion of NO to $NO_2$, which will lead to the accumulation
of $O_3$ concentration.
4.3 Impacts of weakened aerosol-radiation interaction on effectiveness of emission
reduction for $O_3$ air quality
Figure 7 shows the changed summer and winter surface-layer MDA8 $O_3$



concentrations caused by anthropogenic emission reduction from 2013 to 2017 with
($\Delta O_3\_EMI$) and without ($\Delta O_3\_NOARI$) ARI, including the effects of weakened ARI on
the effectiveness of emission reduction for $O_3$ air quality ($\Delta O_3\_\Delta ARI\_EMI$, which is
also equal to $\Delta O_3\_EMI$ minus $\Delta O_3\_NOARI$). Comparing with Fig. 7(a1) and (a2) in
summer and Fig. 7(a4) and (a5) in winter, when the impact of ARI is considered, the
concentrations of MDA8 $O_3$ are increased more than that when ARI is not taken into
account. Thus, $\Delta O_3\_\Delta ARI\_EMI$ makes the superimposed impact on the effectiveness
of anthropogenic emission reduction for the increased MDA8 $O_3$ concentrations from
2013 to 2017 over eastern China. However, during summer, the worsened $O_3$ air quality
due to weakened ARI can only be found in scattered city clusters (e.g., BTH, YRD and
PRD in Fig. 7(a3)). During winter, it would increase MDA8 $O_3$ concentrations over
nearly the whole eastern China (Fig. 7(a6)).

## 5 Conclusions

In this study, the impact of weakened aerosol-radiation interaction (ARI) due to
decreased anthropogenic emissions on surface $O_3$ ($\Delta O_3\_\Delta ARI\_EMI$) over eastern
China is mainly analyzed by using an online-coupled regional chemistry transport
model WRF-Chem. Simulation results generally reproduce the spatiotemporal
characteristics of observations with correlation coefficients of 0.63-0.90 for pollutant
concentrations and 0.40-0.99 for meteorological parameters, respectively.
Sensitivity experiments show that the changes in MDA8 $O_3$ from 2013 to 2017
over eastern China vary spatially and seasonally, and the decreased anthropogenic
emission plays a more prominent role for the MDA8 $O_3$ increase than the impact of
changed meteorological conditions both in summer and winter. Furthermore, the
decreased $PM_{2.5}$ concentrations due to reduced anthropogenic emissions can result in a
weaker impact of ARI on $O_3$ concentrations, which finally pose a superimposed effect
on the worsened $O_3$ air quality. For urban areas over eastern China, $\Delta O_3\_\Delta ARI\_EMI$
has a significant effect on the increase of MDA8 $O_3$ in summer with the value of +1.77
ppb, accounting for 87.6% of the increased value caused by decreased anthropogenic
emissions, but the impacts in winter are smaller (+0.42 ppb), accounting for 11.8% of



the increased value caused by decreased anthropogenic emissions. For the whole
regions over eastern China, the enhancement of MDA8 $O_3$ by $\Delta O_3\_\Delta ARI\_EMI$ is +0.81
(+0.63) ppb, with $\Delta O_3\_\Delta API\_EMI$ and $\Delta O_3\_\Delta ARF\_EMI$ contributing for 55.6%
(61.9%) and 44.4% (38.1%) in summer (winter), respectively. Process analysis shows
that the enhanced $O_3$ chemical production is the dominant process for the increased $O_3$
concentrations caused by $\Delta O_3\_\Delta ARI\_EMI$ both in summer and winter.

Generally, since China's clean air action from 2013, the decreased $PM_{2.5}$

concentrations due to reduced anthropogenic emissions can worsen $O_3$ air quality by
the weakened interactions between aerosol and radiation, which is a new and an
important implication for understanding the causes driving the increases in $O_3$ level
over eastern China. Therefore, our results highlight that more carefully designed multi-
pollutants coordinated emissions control strategies are needed to reduce the
concentrations of $PM_{2.5}$ and $O_3$ simultaneously.



## Acknowledgements

This work is supported by National Natural Science Foundation of China (Grant 42305121, 42007195, 42293320), National Key R&D Program of China (Grant 2019YFA0606804, 2022YFE0136100), Natural Science Foundation of Jiangsu Province (Grant BK20220031), Guizhou Provincial Science and Technology Projects of China (CXTD [2022]001, GCC [2023]026), and Open fund by Jiangsu Key Laboratory of Atmospheric Environment Monitoring and Pollution Control (KHK 2211).



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





**Table 1.** Descriptions of model sensitivity experiments.

| Cases | Anthropogenic emission | Meteorological field | API[a] | ARF[a] |
|---|---|---|---|---|
| **BASE_17E17M** | 2017 | 2017 | On | On |
| **BASE_13E13M** | 2013 | 2013 | On | On |
| **NOAPI_17E17M** | 2017 | 2017 | Off | On |
| **NOALL_17E17M** | 2017 | 2017 | Off | Off |
| **BASE_13E17M** | 2013 | 2017 | On | On |
| **NOAPI_13E17M** | 2013 | 2017 | Off | On |
| **NOALL_13E17M** | 2013 | 2017 | Off | Off |

[a]API means aerosol-photolysis interaction, ARF means aerosol-radiation feedback.

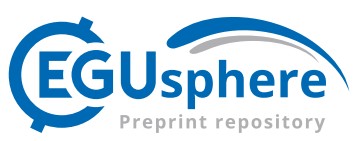

**Table 2.** Statistical parameters of the simulated 2 m temperature ($T_2$, k), 2 m relative humidity ($RH_2$, %), 10 m wind speed ($WS_{10}$, m s$^{-1}$), 10 m wind direction ($WD_{10}$, °), photolysis rate of $NO_2$ ($J[NO_2]$, $10^{-3}$ s$^{-1}$), $PM_{2.5}$ (μg m$^{-3}$), $O_3$ (ppb), and $NO_2$ (ppb) against observations during summer and winter in 2017.

| Variable | Summer | | | | | | Winter | | | | | |
| --- | --- | --- | --- | --- | --- | --- | --- | --- | --- | --- | --- | --- |
| | $O^a$ | $M^a$ | $R^b$ | $MB^c$ | $NMB^d$ (%) | $RMSE^e$ | $O^a$ | $M^a$ | $R^b$ | $MB^c$ | $NMB^d$ (%) | $RMSE^e$ |
| $T_2$ | 295.3 | 294.2 | 0.99 | -1.0 | -3.2 | 1.0 | 275.0 | 272.8 | 0.92 | -2.0 | -74.1 | 2.5 |
| $RH_2$ | 68.1 | 71.0 | 0.97 | 2.2 | 3.2 | 3.6 | 58.1 | 60.6 | 0.87 | 2.1 | 3.5 | 6.5 |
| $WS_{10}$ | 2.6 | 4.2 | 0.77 | 1.6 | 61.6 | 1.6 | 2.6 | 4.7 | 0.82 | 2.1 | 83.2 | 2.1 |
| $WD_{10}$ | 175.7 | 170.9 | 0.40 | -4.6 | -2.6 | 16.9 | 192.6 | 184.6 | 0.69 | -7.5 | -3.9 | 17.4 |
| $J[NO_2]$ | 2.6 | 2.7 | 0.93 | 0.1 | 4.8 | 1.2 | 1.0 | 1.2 | 0.94 | 0.1 | 12.3 | 0.6 |
| $PM_{2.5}$ | 31.0 | 24.8 | 0.63 | -6.3 | -20.2 | 8.3 | 69.0 | 58.9 | 0.80 | -10.1 | -14.6 | 15.6 |
| $O_3$ | 39.7 | 38.9 | 0.90 | -0.6 | -1.6 | 6.9 | 17.7 | 20.5 | 0.86 | 2.8 | 15.7 | 5.0 |
| $NO_2$ | 12.7 | 11.2 | 0.73 | -1.5 | -12.0 | 4.5 | 23.3 | 18.7 | 0.83 | -4.5 | -19.4 | 5.6 |

[a] $O$ and $M$ are the averages for observed and simulated results, respectively. $O = \frac{1}{n} \times \sum_{i=1}^{n} O_i$, $M = \frac{1}{n} \times \sum_{i=1}^{n} M_i$.

[b] $R$ is the correlation coefficient between observations and model results. $R = \frac{\sum_{i=1}^{n}|(O_i - O) \times (M_i - M)|}{\sqrt{\sum_{i=1}^{n}(O_i - O)^2 + \sum_{i=1}^{n}(M_i - M)^2}}$.

[c] $MB$ is the mean bias between observations and model results. $MB = \frac{1}{n} \times \sum_{i=1}^{n}(M_i - O_i)$.

[d] $NMB$ is the normalized mean bias between observations and model results. $NMB = \frac{1}{n} \times \sum_{i=1}^{n} \frac{M_i - O_i}{O_i} \times 100\%$.

[e] $RMSE$ is the root-mean-square error of observations and model results. $RMSE = \sqrt{\frac{1}{n} \times \sum_{i=1}^{n}(M_i - O_i)^2}$.

In the above $O_i$ and $M_i$ are the hourly observed and simulated data, respectively, and n is the total number of hours.



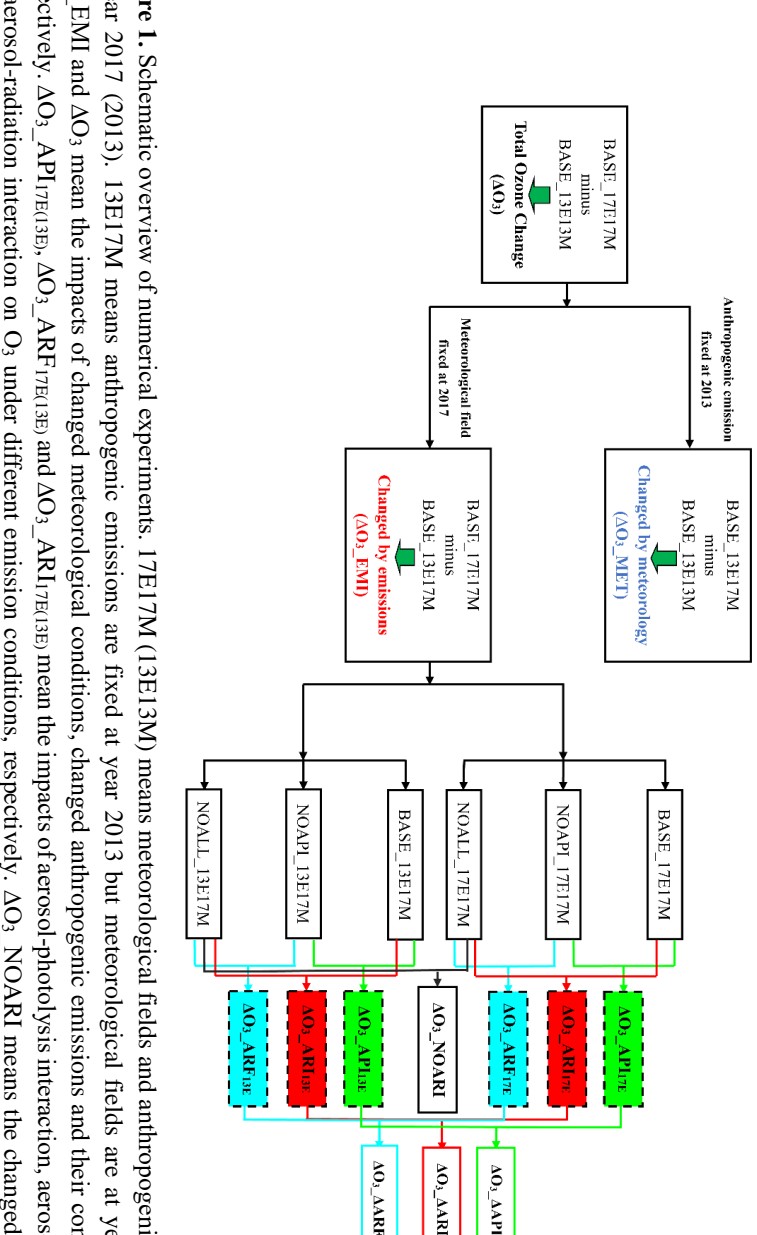

**Figure 1.** Schematic overview of numerical experiments. 17E17M (13E13M) means meteorological fields and anthropogenic emissions are fixed at year 2017 (2013). 13E17M means anthropogenic emissions are fixed at year 2013 but meteorological fields are at year 2017. $\Delta O_3\_MET$, $\Delta O_3\_EMI$ and $\Delta O_3$ mean the impacts of changed meteorological conditions, changed anthropogenic emissions and their combined effects on $O_3$, respectively. $\Delta O_3\_API_{17E(13E)}$, $\Delta O_3\_ARF_{17E(13E)}$ and $\Delta O_3\_ARI_{17E(13E)}$ mean the impacts of aerosol-photolysis interaction, aerosol-radiation feedback and aerosol-radiation interaction on $O_3$ under different emission conditions, respectively. $\Delta O_3\_NOARI$ means the changed $O_3$ concentration by reduced anthropogenic emissions without considering aerosol-radiation interaction. $\Delta O_3\_\Delta API\_EMI$, $\Delta O_3\_\Delta ARF\_EMI$ and $\Delta O_3\_\Delta ARI\_EMI$ represent the impacts of weakened aerosol-photolysis interaction, aerosol-radiation feedback and aerosol-radiation interaction due to decreased anthropogenic emission on $O_3$ concentration, respectively.





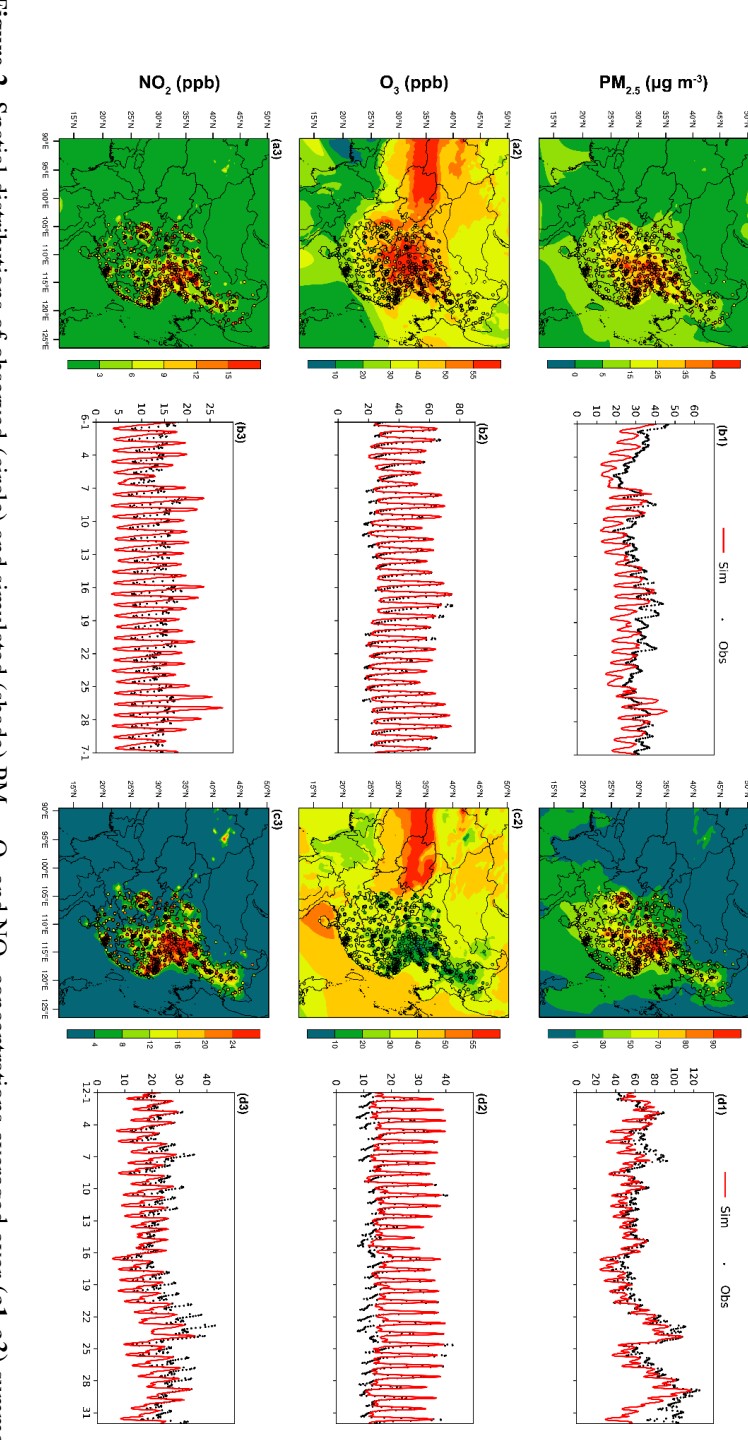

**Figure 2.** Spatial distributions of observed (circle) and simulated (shade) PM$_{2.5}$, O$_3$ and NO$_2$ concentrations averaged over (**a1-a3**) summer and (**c1-c3**) winter in 2017. Time series of observed (black dots) and simulated (red lines) hourly PM$_{2.5}$, O$_3$ and NO$_2$ concentrations averaged over the whole observation sites in eastern China during (**b1-b3**) summer and (**d1-d3**) winter in 2017.





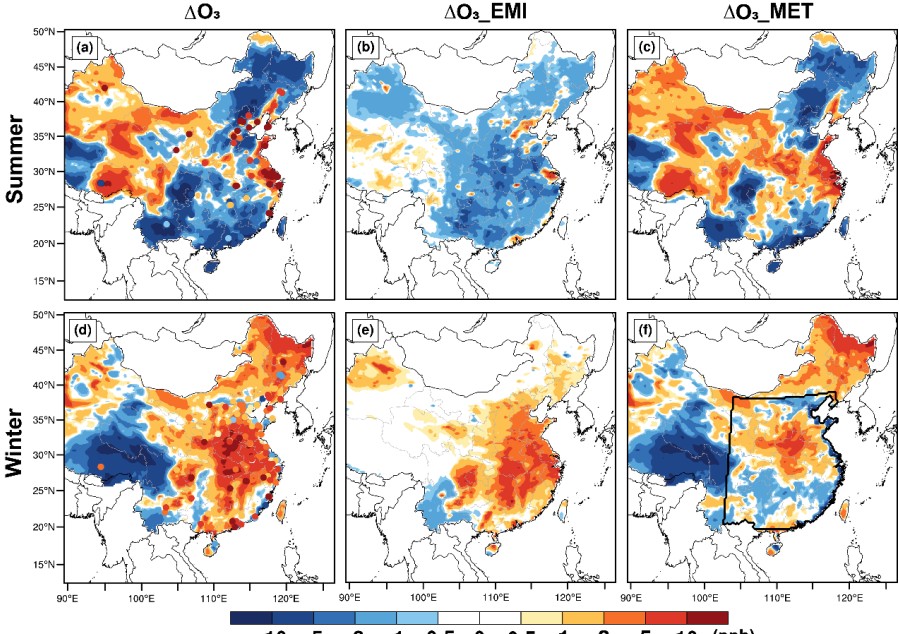

**Figure 3. (a, d)** Spatial distribution of changed summer (upper) and winter (bottom) surface-layer MDA8 O$_3$ from 2013 to 2017, and the contributions of **(b, e)** changed anthropogenic emissions alone and **(c, f)** changed meteorological fields alone. The observed changes in surface MDA8 O$_3$ are also marked with colored circles in **(a)** and **(d)**. The enclosed black line in **(f)** represents eastern China.



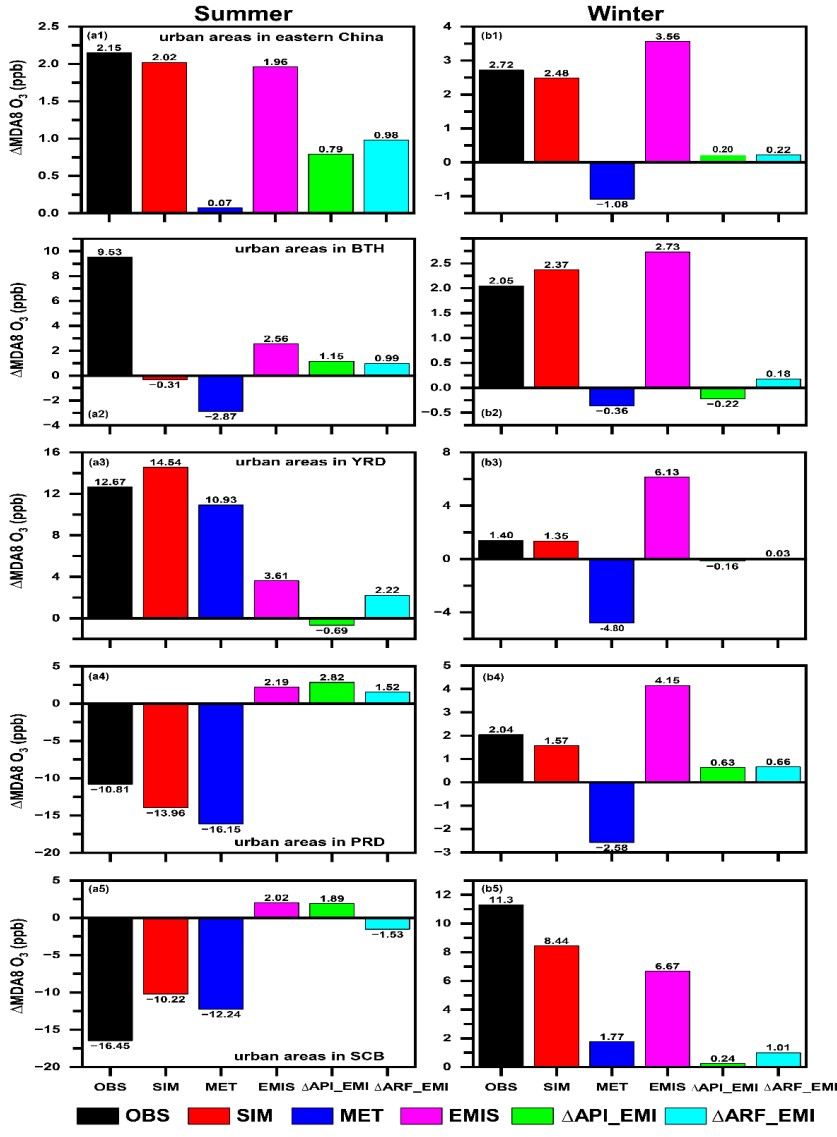

**Figure 4.** The observed (OBS, black bars) and simulated (SIM, red bars) changes in (left) summer and (right) winter surface-layer MDA8 O$_3$ from 2013 to 2017. Contributions of changed meteorological conditions alone (MET, blue bars), changed anthropogenic emissions alone (EMI, purple bars), changed aerosol-photolysis interaction alone (ΔAPI_EMI, green bars), and changed aerosol-radiation feedback alone (ΔARF_EMI, cyan bars) are also shown. Observations are calculated from the monitoring sites in the analyzed region, while the corresponding gridded simulations are averaged for SIM. **(a1-b1)**, **(a2-b2)**, **(a3-b3)**, **(a4-b4)** and **(a5-b5)** represent the urban areas in eastern China, Beijing-Tianjin-Hebei (BTH), Yangtze River Delta (YRD), Pearl River Delta (PRD), and Sichuan Basin (SCB), respectively.





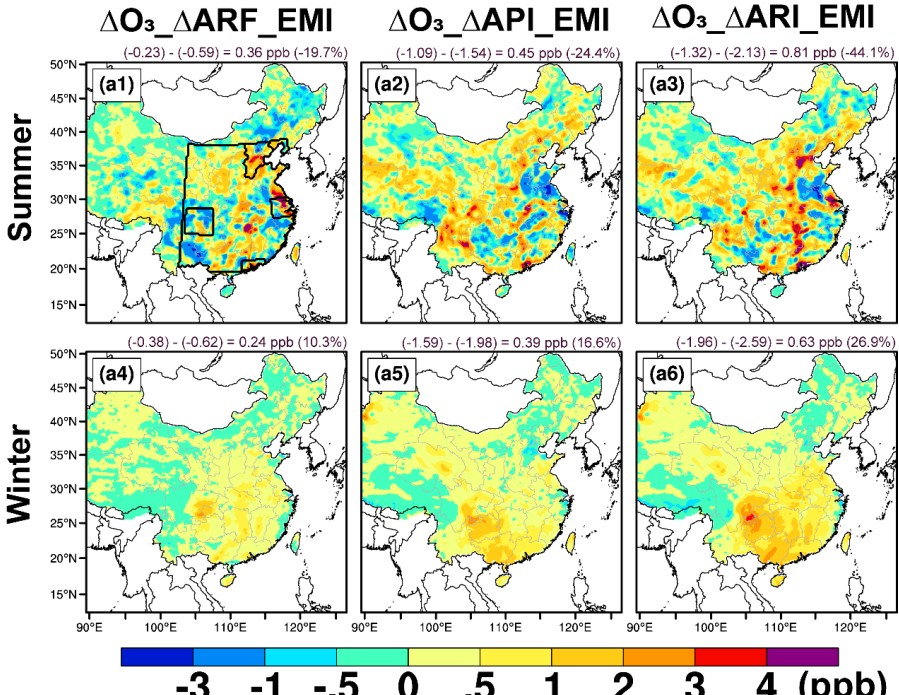

**Figure 5.** Impacts of $\Delta O_3\_\Delta ARF\_EMI$, $\Delta O_3\_\Delta API\_EMI$, and $\Delta O_3\_\Delta ARI\_EMI$ on summer (upper) and winter (bottom) surface-layer MDA8 $O_3$ concentrations. The enclosed black line in **(a1)** represents eastern China and the four developed city clusters. The mean changes over eastern China are also shown at the top of each panel. Detailed information about $\Delta O_3\_\Delta ARF\_EMI$, $\Delta O_3\_\Delta API\_EMI$, and $\Delta O_3\_\Delta ARI\_EMI$ can be found in Figure 1.

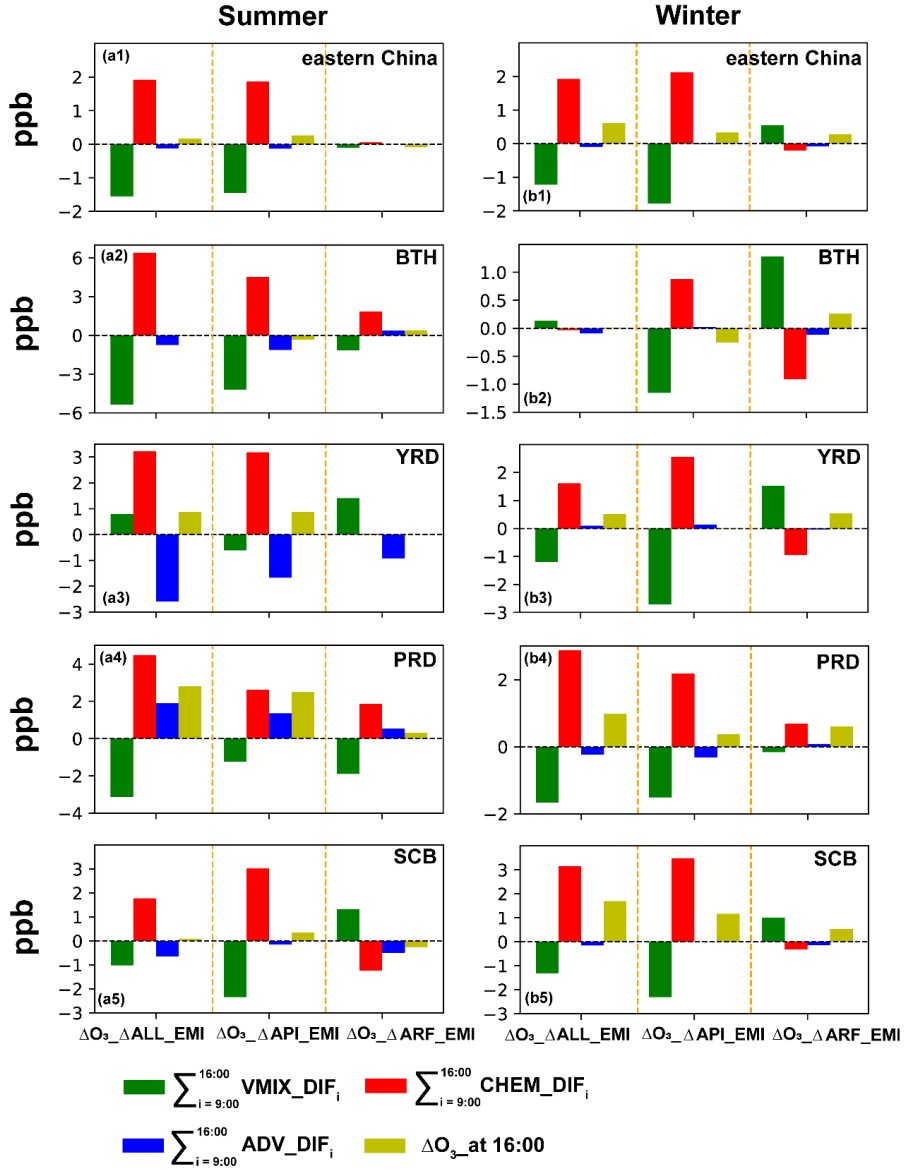

**Figure 6.** Accumulated changes in each process from 09:00 to 16:00 LST and the changed $O_3$ concentrations due to $\Delta O_3\_\Delta ARI\_EMI$ in summer (left column) and winter (right column). The regions of eastern China, Beijing-Tianjin-Hebei (BTH), Yangtze River Delta (YRD), Pearl River Delta (PRD) and Sichuan Basin (SCB) are indicated on the right side of each panel.



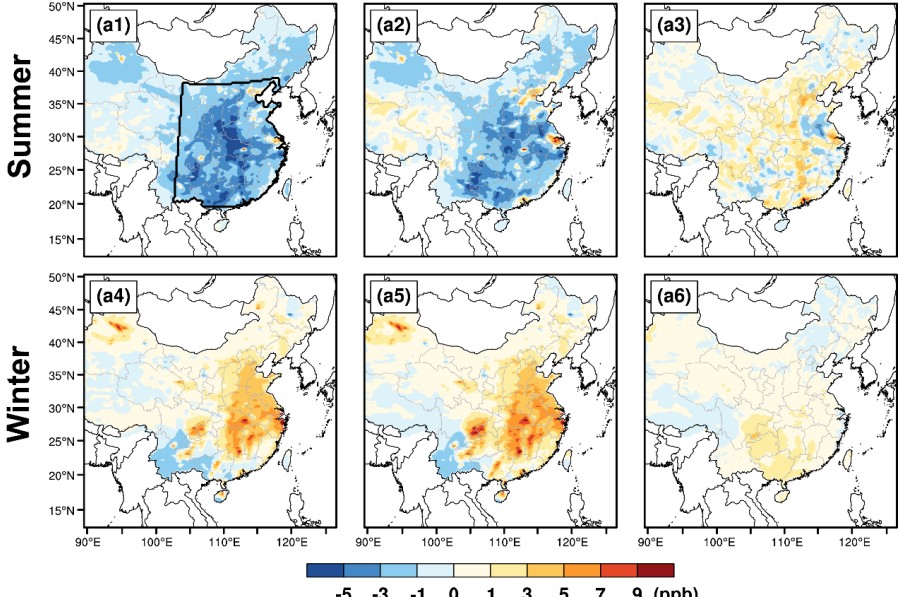

**Figure 7.** Spatial distribution of changed summer (upper) and winter (bottom) surface-layer MDA8 $O_3$ concentrations from sensitivity simulations. **(a1, a4)** Effects of anthropogenic emission reduction on MDA8 $O_3$ without ARI. **(a2, a5)** Effects of anthropogenic emission reduction on MDA8 $O_3$ with ARI. **(a3, a6)** Effects of weakened ARI on the effectiveness of emission reduction for $O_3$ air quality.