# Peer review of "Weakened aerosol-radiation interaction exacerbating ozone"

_EGUsphere, 2023_

## Author Comment (AC1)

**Response to Comments of Reviewer #1**

**(comments in *italics*)**

**Manuscript number:** EGUSPHERE-2023-2393

**Title:** Weakened aerosol-radiation interaction exacerbating ozone pollution in eastern China since China's clean air actions

*This paper mainly investigated the impacts of aerosol-photolysis interaction (API) and aerosol-radiation feedback (ARF) on the surface ozone concentrations under the background of China's clean air action (rapid anthropogenic emission reductions from 2013 to 2017).*

*The effects of API on ozone concentrations are not a new finding since I have found several previous studies already addressed it (Gao et al., 2022; Liu and Wang, 2020). However, I have not found any previous studies focused on the effects of ARF on ozone concentrations. Furthermore, the authors used the IPR methodology to investigate the contribution to O₃ concentration variation from four processes (VMIX, CHEM, ADVH, ADVZ). In conclusion, I consider this paper valuable for publication, even if it has some limitations (as shown below). (1) The absence of SOA formation and heterogeneous reactions in their simulations could be a limitation of this study; even the authors have sufficiently acknowledged this. (2) Some parts/aspects are poorly elucidated, making it hard for me to understand. A major revision is needed before it can be published in ACP.*

**Response:**

Thanks to the reviewer for the valuable comments and suggestions which are very helpful for us to improve our manuscript. We have revised the manuscript carefully, as described in our point-to-point responses to the comments.

The major innovation of this study is that **it is the first time** to quantify the response of aerosol-radiation interaction to anthropogenic emission reduction from 2013 to 2017, with the mainly focus on the contribution to changed $O_3$ concentrations over eastern China both in summer and winter.

According to the reviewer's comments, **another three** widely used chemical mechanisms, i.e., RADM2-MADE/SORGAM (RADM2 gas-phase chemistry coupled with MADE/SORGAM aerosol module), CBMZ-MADE/SORGAM (CBMZ gas-phase chemistry coupled with MADE/SORGAM aerosol module), and MOZART-MOSAIC (MOZART gas-phase chemistry coupled with MOSAIC aerosol module), that include SOA formation are also applied to assess the impact of aerosol-radiation interaction (ARI) on $O_3$.

Comparing the simulation results of the three additional mechanisms, the simulated $PM_{2.5}$ from MOZART-MOSAIC are closer to the actual observation. Analyzing the summer/winter MDA8 $O_3$ reductions due to ARI by the mechanism used in our manuscript (i.e., CBMZ-MOSAIC) and MOZART-MOSAIC, **similar results are quantified** (1.32 ppb *vs.* 1.85 ppb for summer, and 1.96 ppb *vs.* 1.60 ppb for winter). Therefore, although the CBMZ-MOSAIC used in this paper does not take into account the formation of SOA and its associated effects, the aerosol radiative effect on $O_3$ concentration is consistent with the results when the SOA simulation mechanism is considered.

The impacts of aerosol heterogeneous reactions on $O_3$ have not been considered in this

manuscript due to the **uncertainty and inconsistency** of the heterogeneous uptake shown in previous observation and simulation studies (Liu and Wang., 2020b; Tan et al., 2020; Shao et al., 2021). Shao et al. (2021) summarized that different heterogeneous uptake on the aerosol surface applied in the model simulation (e.g., 0.20 *vs.* 0.08) would cause significant deviations in simulated ozone concentrations (e.g., $O_3$ increase by 6% *vs.* $O_3$ increase by 2.5%). Therefore, the uncertainty in the heterogeneous uptake value used in the numerical simulation will finally amplify the deviation in model results.

According to the reviewer's comments about some poorly elucidated parts, such as $\Delta O_3\_\Delta ARF\_EMI$. We have **detailedly described** in our point-to-point responses as shown below, and related descriptions have also been added in the revised manuscript.

**Specific comments:**

1. *In my opinion, SOAs account for a substantial portion of total aerosols. Typically, in your research, the lack of consideration of SOA can truly affect the reliability of the results (the authors also mentioned that $PM_{2.5}$ is underestimated in your model). I highly recommend the authors include SOA formation in their model.*

**Response:**

Thanks to the reviewer for the valuable comments and suggestions. The CBMZ gas-phase chemistry coupled with MOSAIC aerosol module (CBMZ-MOSAIC for short) used in this study does not include secondary organic aerosol (SOA), then we applied three additional chemical mechanisms that consider SOA, namely, RADM2 gas-phase chemistry coupled with MADE/SORGAM aerosol module (RADM2-MADE/SORGAM for short), CBMZ gas-phase chemistry coupled with MADE/SORGAM aerosol module (CBMZ-MADE/SORGAM for short), and MOZART gas-phase chemistry coupled with MOSAIC aerosol module (MOZART-MOSAIC for short), to test the impact of ARI on $O_3$ with and without SOA for the scenario of BASE_17E17M.

Figures R1 shows the temporal variations of observed and simulated $PM_{2.5}$ and $O_3$ concentrations over eastern China for the three additional chemical mechanisms. Comparing with the observed $PM_{2.5}$ ($O_3$) concentrations, the MOZART-MOSAIC showed the best performance in December 2017, with the R of 0.73 (0.79) and NMB of -18.7% (-20.5%). Therefore, we further used this mechanism to simulate the air pollutant concentrations during the period of June 2017. As shown in Fig. R1 (a4, b4), the temporal variations of observed $PM_{2.5}$ ($O_3$) can be well captured by this mechanism with R of 0.56 (0.91) and NMB of -1.7% (-20.3%).

Finally, we investigated the effect of aerosol-radiation interaction (ARI) on $O_3$ from the results of CBMZ-MOSAIC (this mechanism applied in this manuscript which does not include SOA) and MOZART-MOSAIC (this mechanism includes SOA and performs the best simulation results comparing with RADM2-MADE/SORGAM and CBMZ-MADE/SORGAM). As shown in Fig. R2, summer (winter) MDA8 $O_3$ is significantly reduced over eastern China, ARI reduces the surface MDA8 $O_3$ concentrations by 1.32 (1.96) ppb and 1.85 (1.60) ppb by CBMZ-MOSAIC and MOZART-MOSAIC, respectively. The $O_3$ reductions are of comparable magnitude in these two schemes. Therefore, we can conclude that although the CBMZ-MOSAIC applied in this manuscript does not take into account the formation of SOA and its associated effects, the aerosol radiative effects on $O_3$ concentrations not only in the pattern of spatial-temporal distribution but also in the order of magnitude are consistent with the results when the SOA simulation mechanism is considered.

As shown in Fig. R3, the mean SOA simulated by RADM2-MADE/SORGAM, CBMZ-

MADE/SORGAM, and MOZART-MOSAIC are 0.29, 0.45 and 0.94 µg m$^{-3}$, accounting for 3.4%, 3.8%, and 4.4% of PM$_{2.5}$ concentrations in winter 2017, respectively. From Fig. R4, the mean SOA simulated from MOZART-MOSAIC is 0.90 µg m$^{-3}$, account for 9.1% of PM$_{2.5}$ in summer 2017. Model simulated SOA concentrations are generally underestimated in most current chemical transport models (Zhang et al., 2015; Zhao et al., 2015). The low SOA concentrations simulated by the model can be explained by low emissions of biogenic and anthropogenic VOCs (key precursors of SOA), but a thorough investigation of this underestimation is outside the scope of this manuscript and it will be discussed in our future work. (**Page 18-19, Line 497-536**)

According to the reviewer's suggestion, we have added Figs. R1-R4 in the revised support information. **(Page 13-16 in supporting information)**

[Figure]

**Figure R1.** Time series of observed (black dots) and simulated (red lines) hourly (a1-a4) PM$_{2.5}$ and (b1-b4) O$_3$ concentrations averaged over the whole observation sites in eastern China during summer and winter 2017. (a1, b1) Simulated PM$_{2.5}$ and O$_3$ concentrations in winter 2017 by CBMZ gas-phase chemistry coupled with MOSAIC aerosol module (RADM2-MADE/SORGAM). (a2, b2) Simulated PM$_{2.5}$ and O$_3$ concentrations in winter 2017 by CBMZ gas-phase chemistry coupled with MADE/SORGAM aerosol module (CBMZ-MADE/SORGAM). (a3, b3) Simulated PM$_{2.5}$ and O$_3$ concentrations in winter 2017 by MOZART gas-phase chemistry coupled with MOSAIC aerosol module (MOZART-MOSAIC). (a4, b4) is the same as (a3, b3), but for summer 2017. The calculated correlation coefficient (R), mean bias (MB), and normalized mean bias (NMB) are also shown.

[Figure]

**Figure R2.** The effects of aerosol-radiation interaction on surface-layer MDA8 O₃ in summer (upper) and winter (bottom) 2017 calculated by **(a, c)** CBMZ-MOSAIC and **(b, d)** MOZART-MOSAIC chemical mechanisms. The changes (percentage changes) averaged over China are also shown at the top of each panel.

[Figure]

**Figure R3.** Spatial distributions of simulated mean PM₂.₅ and SOA concentrations (μg m⁻³) in winter 2017 by **(a)** CBMZ gas-phase chemistry coupled with MOSAIC aerosol module (CBMZ-MOSAIC), **(b, e)** RADM2 gas-phase chemistry coupled with MADE/SORGAM aerosol module (RADM2-MADE/SORGAM), **(c, f)** CBMZ gas-phase chemistry coupled with MADE/SORGAM aerosol module (CBMZ-MADE/SORGAM), and **(d, g)** MOZART gas-phase chemistry coupled with MOSAIC aerosol module (MOZART-MOSAIC). The calculated pollutant concentrations averaged over China are also shown at the top of each panel.

[Figure]

**Figure R4.** Spatial distributions of simulated mean PM$_{2.5}$ and SOA concentrations ($\mu$g m$^{-3}$) in summer 2017 by **(a)** CBMZ gas-phase chemistry coupled with MOSAIC aerosol module (CBMZ-MOSAIC), (b, c) MOZART gas-phase chemistry coupled with MOSAIC aerosol module (MOZART-MOSAIC). The calculated pollutant concentrations averaged over China are also shown at the top of each panel.

2. *Similarly, as the significant impacts of heterogeneous reactions on ozone concentrations mentioned by previous studies (Lou et al., 2014; Liu and Wang, 2020), I would expect the authors to include heterogeneous reactions in their models. If the authors have specific reasons for not including heterogeneous reactions in their models, those reasons need to be stated in the paper.*

**Response:**

In addition to the impacts of aerosol-radiation interaction (ARI), aerosols can also affect the concentrations of O$_3$ by heterogeneous chemistry (HET). Liu and Wang. (2020b) found that the rapid decrease of PM$_{2.5}$ was a major contributor for the summer O$_3$ increase through weakening the heterogeneous uptake of hydroperoxy radical (HO$_2$). However, Tan et al. (2020) launched a field campaign in North China Plain (NCP) and proposed a contradicting opinion about the importance of the impact of HET on O$_3$. These inconsistent conclusions generated from field observations and numerical simulations are mainly originated from the different values of heterogeneous uptake they used. Tan et al. (2020) pointed out that the heterogeneous uptake of HO$_2$ on aerosol surface was 0.08 ($\gamma_{HO2} = 0.08$) over NCP, which is smaller than the values ($\gamma_{HO2} = 0.2$) used in model simulations (Li et al., 2019; Liu and Wang., 2020). As shown in Fig. R5, Shao et al. (2021) found controversial results by using the different heterogeneous uptake of HO$_2$. When $\gamma_{HO2} = 0.2$ was used in the chemical model, the reduced heterogeneous uptake of HO$_2$ due to the decrease in aerosol caused the maximum O$_3$ increased by about 6% from 2013 to 2016, which is close to the results of Li et al. (2019) (~ 7%). When $\gamma_{HO2} = 0.08$ was used, the reduced heterogeneous uptake of HO$_2$ due to the decrease in aerosol led to maximum O$_3$ increased by only 2.5% from 2013 to 2016. Therefore, significant deviations in the model results would result from the use of different heterogeneous uptake on the aerosol surface.

Furthermore, previous laboratory studies indicate that the uptake coefficient varies widely from 0.003 to 0.5 with a strong dependence on the concentration of transition metal ions such as Cu(II) and Fe(II) in the aerosol (Zou et al., 2019). Taketani et al. (2009) reported that the uptake coefficient of HO$_2$ ($\gamma_{HO2}$) on seawater particles depends on relative humidity (RH), with $\gamma_{HO2}$ values of 0.10 $\pm$

0.03, 0.11 ± 0.02 and 0.10 ± 0.03 at 35%, 50% and 75% RH, respectively. Lakey et al. (2015) also found that a large humidity dependence was observed for HO$_2$ uptake onto humic acid aerosols. The HO$_2$ uptake coefficient increased from 0.007 ± 0.002 to 0.06 ± 0.01 between 32 and 76% RH for the Acros organics humic acid, and from 0.043 ± 0.009 to 0.09 ± 0.03 between 33 and 75% RH for the Leonardite humic acid. This strong dependence on aerosol composition and RH implies that a single assumed value for heterogeneous uptake used in numerical simulation may cause large uncertainty. In addition, our manuscript devoted to quantifying the effects of ARI on O$_3$, rather than the impacts of heterogeneous reactions on O$_3$. Due to the reasons listed above, we did not consider the effect of heterogeneous reactions on O$_3$ temporarily in the manuscript.

Thanks for the reviewer's suggestion, and we will consider the impacts of heterogeneous reaction in our future works. A discussion about the impacts of heterogeneous reaction has been added in the revised manuscript as follows:

"The impacts of aerosol heterogeneous reactions (HET) on O$_3$ have not been considered in this manuscript due to the uncertainty and inconsistency of the heterogeneous uptake shown in previous observation and simulation studies (Liu and Wang., 2020b; Tan et al., 2020; Shao et al., 2021). Liu and Wang. (2020b) found that the rapid decrease of PM$_{2.5}$ was the primary contributor for the summer O$_3$ increase through weakening the heterogeneous uptake of hydroperoxy radical (HO$_2$). However, Tan et al. (2020) launched a field campaign in NCP and proposed a contradicting opinion about the importance of the impact of HET on O$_3$. Shao et al. (2021) summarized that different heterogeneous uptake on the aerosol surface applied in the model simulation (e.g., 0.20 *vs.* 0.08) would cause significant deviations in simulated ozone concentrations (e.g., O$_3$ increase by 6% *vs.* O$_3$ increase by 2.5%). Previous laboratory studies indicate that the dependence of the uptake coefficient on aerosol composition and RH means that a single assumed value for heterogeneous uptake used in numerical simulations can lead to large uncertainties (Lakey et al., 2015; Taketani et al., 2009; Zou et al., 2019). Therefore, the uncertainty in the heterogeneous uptake value used in the numerical simulation will finally amplify the deviation in model results. Meanwhile, our manuscript devoted to quantifying the effects of ARI on O$_3$, rather than the impacts of heterogeneous reactions on O$_3$. The absence of heterogeneous chemistry on aerosol surface may result in underestimation of the effect of aerosol on O$_3$, which will be considered in our future work." (**Page 19-20, Line 537-556**)

[Figure]

**Figure R5.** O$_3$ change due to the decrease in PM$_{2.5}$ during 2006-2016 and during 2013-2016 in the study of Shao et al., (2021) and during 2013-2017 in the study of Li et al., (2019a). This picture is from Shao et al., (2021).

3. *L160, you mentioned you fixed the meteorological field to the year 2013, can you explain how to achieve this? Can I understand that all \*17M cases have exactly the same meteorological fields throughout 2017 simulation? However, I don't think all \*17M cases should have the same meteorological fields, because you cannot investigate deltaO3_deltaARF_EMI if the meteorological fields are fixed in different cases. This needs to be explained more clearly in your paper.*

**Response:**

Thanks for your comments. $\Delta O_3\_\Delta ARF\_EMI$ represents the impacts of weakened aerosol-radiation feedback ($\Delta ARF$) due to decreased anthropogenic emission (EMI) on $O_3$ concentrations ($\Delta O_3$). In order to quantify the impacts caused by the decreased EMI from 2013 to 2017, the impacts of changed meteorological variables should be removed by fixing the meteorological fields in year 2017 in sensitivity experiments, such as NOAPI_13E17M, NOALL_13E17M, NOAPI_17E17M and NOALL_17E17M (13E17M means anthropogenic emissions are from the year of 2013 and meteorological fields are from the year of 2017, more details can be found in Figure 1 in the revised manuscript).

For example, the differences between NOAPI_13E17M and NOALL_13E17M reflect the impact of ARF at the condition of 13E17M (the result is denoted as $\Delta O_3\_ARF_{13E}$ for short), and the differences of NOAPI_17E17M and NOALL_17E17M show the impact of ARF at the condition of 17E17M (the result is denoted as $\Delta O3\_ARF_{17E}$ for short), so the differences between $\Delta O_3\_ARF_{17E}$ and $\Delta O3\_ARF_{13E}$ finally present the impact of weakened aerosol-radiation feedback due to decreased anthropogenic emission from 2013 to 2017 on $O_3$ concentrations.

For the summer simulations and the winter simulation in the year of 2013 or in the year of 2017, we use the June and December meteorological fields for the corresponding year.

The same method has been widely used in many other studies, which mainly focus on the impacts of weakened aerosol-radiation interactions on air pollutants in China (Li et al. 2019; Zhou et al., 2019; Hong et al. 2020; Liu and Wang. 2020b; Zhu et al. 2021; Shao et al. 2021).

According to the reviewer's suggestion, we have added this information in the revised manuscript. (**Page 7-8, Line 175-214**)

4. *L23-L25, you mentioned API and ARF. However, the API and ARF terminology is so abstract, making it hard for people to understand. It would help if you mentioned that API is related to the change in photolysis rates and ARF is related to the change of meteorological fields in your abstract.*

**Response:**

Thanks for your suggestion, we have added this information in the revised manuscript as follows: **"**Here we apply a coupled meteorology-chemistry model (WRF-Chem) to quantify the responses of aerosol-radiation interaction (ARI), including aerosol-photolysis interaction (API) related to photolysis rate change and aerosol-radiation feedback (ARF) related to meteorological fields change, to anthropogenic emission reductions from 2013 to 2017, and their contributions to $O_3$ increases over eastern China in summer and winter.**"** (**Page 2, Line 24-26**)

5. *L58, I think chemical species like CO and $CH_4$ can also lead to the formation of $O_3$.*

**Response:**

According to the reviewer's suggestion, we have changed the sentence in the revised

manuscript as follows: "As a secondary air pollutant, troposphere $O_3$ can be produced by nitrogen oxides ($NO_x$ = NO + $NO_2$), carbon monoxide (CO), methane ($CH_4$) and volatile organic compounds (VOCs) in the presence of solar radiation through photochemical reactions (Atkinson, 2000; Seinfeld and Pandis, 2006)." (**Page 3, Line 60**)

*6. L57-L62 The causal relationship between the following two sentences is not clear. As a secondary air pollutant, troposphere O₃ can be produced by nitrogen oxides (NOₓ = NO + NO₂) and volatile organic compounds (VOCs) in the presence of solar radiation through photochemical reactions (Atkinson, 2000; Seinfeld and Pandis, 2006). - > Consequently, the concentration of O₃ is closely related to changes in meteorological conditions and anthropogenic emissions (Wang et al., 2019; Liu and Wang, 2020a,b; Shu et al., 2020). "solar radiation" is not directly related to "meteorological conditions", try to revise those sentences to make them more logical.*
**Response:**

Thanks for your suggestion. we have changed the sentence in the revised manuscript as follows: "The concentration of $O_3$ in the troposphere is influenced by changes in meteorological conditions (e.g., high temperature and low relative humidity) and its precursors emissions (e.g., NOx and VOCs) (Wang et al., 2019; Liu and Wang, 2020a,b; Shu et al., 2020). Most precursors are from anthropogenic sources, and some precursors can come from natural sources, such as biogenic VOCs and soil and lightning NOx." (**Page 3, Line 62-67**)

*7. 2.1 Model configuration: I recommend using a chart (like Table 1 in https://www.sciencedirect.com/science/article/pii/S1352231020307378) to summarize the model configuration.*
**Response:**

Thanks to the reviewer's comments, the model configuration is summarized in Table R1. We have added Table R1 in the revised supporting information. (**Table S1**)

**Table R1.** WRF-Chem model configurations with main physical and chemical schemes adopted in this study.

| Model set-up | Values |
| --- | --- |
| Domain | East Asia |
| Study period | June and December 2017 |
| Domain size | 167 × 167 |
| Domain center | 34 °N, 108 °E |
| Horizontal resolution | 27 km × 27 km |
| Vertical resolution | 32 eta levels up to 50 hPa |
| Meteorological boundary and initial conditions | NCEP 1°×1° reanalysis data |
| Chemical initial and boundary conditions | CAM-Chem output |
| **Physical options** | **Adopted scheme** |
| Microphysics scheme | Lin (Purdue) scheme |
| Cumulus scheme | Grell 3D ensemble scheme |
| Boundary layer scheme | Yonsei University PBL scheme |
| Surface layer scheme | Monin-Obukhov surface scheme |
| Land-surface scheme | Unified Noah land-surface model |

| Longwave radiation scheme | RRTMG |
|---|---|
| Shortwave radiation scheme | RRTMG |
| **Chemical options** | **Adopted scheme** |
| Gas phase chemistry | CBMZ |
| Aerosols | MOSAIC |
| Photolysis | Fast-J |
| Biogenic emissions | MEGAN |
| Anthropogenic emissions | MEIC |

8. *L125-L127, have you applied meteorological nudging? See above, I am not sure how you fix the meteorological fields to 2013 or 2017 when running the model.*

**Response:**

This work is done without nudging because only one domain is designed in our manuscript. If the nudging is turned on in only one domain simulation, the simulated meteorological field can not truly reflect the influence of the aerosol-radiation interaction feedback.

When using the 2013 (2017) FNL meteorological field data, it means that the meteorological field are from the year of 2013 (2017). For example, BASE_17E17M means that the meteorological field and anthropogenic emission are from the year of 2017. BASE_13E13M means that the meteorological field and anthropogenic emission are from the year of 2013.

9. *L151, you mentioned the biogenic emissions are calculated online by MEGAN. Have you coupled the MEGAN model with WRF-Chem dynamically? Please ascertain whether the biogenic emissions are calculated online or offline by MEGAN.*

**Response:**

Thanks to the reviewer's comments. In this work, we set "bio_emiss_opt = 3" in the WRF-Chem model, which represents the biogenic emissions can be calculated online by the coupled MEGAN module based upon the simulated meteorological variables (e.g., temperature, solar radiation) and underlying static data (e.g., leaf area index, plant types).

10. *L166, can you explain which aerosol optical properties are turned to zero?*

**Response:**

Following Qiu et al. (2017), the aerosol radiation interactions were turned off by removing the mass of aerosol species from the calculation of aerosol optical properties. Then, the aerosol optical properties such as aerosol optical depth (AOD), aerosol single scattering albedo (SSA), aerosol asymmetery factor (g) and aerosol backscatter coefficient were set to zero.

11. *L200-202, you mentioned "To avoid potential deviations caused by long-term model integration, each simulation is re-initialized every eight days". I was confused about why re-initialize the simulation every eight days can avoid potential deviations. What do you mean "potential deviations"? Can you explain this more?*

**Response:**

Thanks to the reviewer's comments. Lo et al. (2008) conducted three types of experiments for the entire year of 2000 to test model performance for different simulation durations: (1) continuous

integrations with a single initialization as usually done, namely, one year of uninterrupted simulation (WRFS), (2) consecutive integrations with re-initializations every 29 days (WRFM-30D), and (3) same as (2) but the model is reinitialized every 6 days (WRFM-7D). They found that the traditional continuous integration approach (WRFS) shows the worst performance. The model drifts from the forcing FNL reanalysis during the course of long integrations. It poorly simulates not only the forcing variables, (e.g., pressure, temperature, wind, and moisture), but also the model diagnostics variables (e.g., precipitation). Therefore, the simulation is re-initialized every eight days in this work, the same as the WRFM-7D, to avoid the deviation from forcing variables, (e.g., pressure, temperature, wind, and moisture) and model diagnostics variables (e.g., precipitation).

*12. L214-217 I feel confused about how many sites are operated by China National Environmental Monitoring Center (CNEMC)? You mentioned "1296 sites", does this number refer to the number of total sites of CNEMC or the number of sites chosen in your research? Moreover, are there really 1296 points (sites) on Figs. 2a and 2c?*

**Response:**

The CNEMC had 1484 observation sites in 2017. In this work, a single site with at least 500 actual observations during the simulated period are used for model evaluation, as we mentioned in the manuscript (**Page 9, Line 238-240**). Of course, Figs. 2a and 2c does have 1296 sites.

*13. Figure 2 shows the simulated results of which case? (BASE_17E17M?) You need to specify this point in L251 and Fig. 2.*

**Response:**

According to the reviewer's comment, we made it clear in Section 3 in the revised manuscript that the simulation results from the case of BASE_17E17M are used to evaluate the model performs (**Page 10, Line 264-266**).

*14. Why there are less points on Figs. 3a and 3d than Fig. 2? Please explain.*

**Response:**

Thanks to the reviewer's comments. The CNEMC installed only 450 sites in 2013, which grew to more than 1500 stations by 2020. In Fig. 3, only sites with continuous observations and individual site data greater than 500 were used to assess ozone trends. Thus, Fig. 3 has fewer points than Fig. 2.

*15. L221, if possible, I recommend explaining more about IPR in your paper.*

**Response:**

Thanks to the reviewer's suggestion, we have added this sentence in the revised manuscript as follows: "Process analysis techniques, i.e., integrated process rate (IPR) analysis, can be used in grid-based Eulerian models (e.g., WRF-Chem) to obtain contributions of each physical/chemical process to variations in pollutant concentrations. Eulerian models utilize the numerical technique of operator splitting to solve continuity equations for each species into several simple ordinary differential equations or partial differential equations that only contain the influence of one or two processes (Gipson, 1999).

In order to quantitatively elucidate individual contributions of physical and chemical processes

to $O_3$ concentration changes due to weakened ARI, the integrated process rate (IPR) methodology is applied in this study. IPR analysis is an advanced tool to evaluate the key process for $O_3$ concentration variation (Shu et al., 2016; Zhu et al., 2021; Yang et al., 2022). In this study, the IPR analysis tracks hourly (e.g., one time step) contribution to $O_3$ concentration variation from four main processes, including vertical mixing (VMIX), net chemical production (CHEM), horizontal advection (ADVH), and vertical advection (ADVZ). VMIX is initiated by turbulent process and closely related to PBL development, which influences $O_3$ vertical gradients. CHEM represents the net $O_3$ chemical production (chemical production minus chemical consumption). ADVH and ADVZ represent transport by winds. We define ADV as the sum of ADVH and ADVZ." (**Page 9-10, Line 245-262**)

*16. Table 2, how many sites are used for Table 2 (1296 sites?)?*
**Response:**
Thanks to the reviewer's comments. Table 2 contains 1296 sites, and we added this information to the revised manuscript (**Page 31, Line 811**).

*17. L284-285, you mentioned NOx-limited and VOCs-limited regions, I recommend that you could add a figure (based on your simulation results) like Fig. 5 in https://www.sciencedirect.com/science/article/pii/S1352231013000514 to your supplement, to show different $O_3$-sensitive regions on the map.*
**Response:**
The typical VOCs/$NO_x$ ratio is calculated to classify sensitivity regimes and to indicate the possible $O_3$ responses to changes in VOCs and/or $NO_x$ concentrations. $O_3$ production is VOC-limited if the ratio is less than 4, and it is $NO_x$-limited if the ratio is larger than 15 (Edson et al., 2017; Li et al., 2017). The ratio of VOCs/$NO_x$ ranging around 4-15 indicates a transitional regime, where ozone is nearly equally sensitive to each species (Sillman, 1999). As shown in Fig R6, $O_3$ are mainly formed under the VOC-limited in winter and $NO_x$-limited and transitional regimes in eastern China, which is consistent with what our study mentioned.

According to the reviewer's suggestion, we have added Fig. R6 in the revised support information. (**Page 7 in supporting information**)

[Figure]

**Figure R6.** The ratios of VOCs/NO$_x$ calculated from (a, b) BASE_17E17M, and (c, d) BASE_13E13M during the daytime (08:00-17:00 LST) from summer (left) and winter (right).

18. *L290-292, the meteorological effects are comparable or larger or smaller than emissions effects? This should be mentioned.*

**Response:**

From Figs. R7, compared with 2013, the meteorological conditions in the summer of 2017 promoted the generation of O$_3$ in the YRD region, but suppressed the generation of O$_3$ in the BTH, PRD and SCB regions. In PRD and SCB, the changes in MDA8 O$_3$ due to meteorology even have a greater impact than that by emission changes, which highlights the significant role of meteorology on summer O$_3$ variations during summer.

Thanks for reviewer's suggestion, we have added this information in the revised manuscript. **(Page 13, Line 343-349)**

[Figure]

**Figure R7.** The observed (OBS, black bars) and simulated (SIM, red bars) changes in (left) summer and (right) winter surface-layer MDA8 $O_3$ from 2013 to 2017. Contributions of changed meteorological conditions alone (MET, blue bars), changed anthropogenic emissions alone (EMI, purple bars), changed aerosol-photolysis interaction alone ($\Delta$API_EMI, green bars), and changed aerosol-radiation feedback alone ($\Delta$ARF_EMI, cyan bars) are also shown. Observations are calculated from the monitoring sites in the analyzed region, while the corresponding gridded simulations are averaged for SIM. **(a1-b1)**, **(a2-b2)**, **(a3-b3)**, **(a4-b4)** and **(a5-b5)** represent the urban areas in eastern China, Beijing-Tianjin-Hebei (BTH), Yangtze River Delta (YRD), Pearl River Delta (PRD), and Sichuan Basin (SCB), respectively.

19. *L429-431, you mentioned "multi-pollutants coordinated emissions control strategies", can you specify this and give more details? Liu and Wang, 2020 suggested that "to reduce $O_3$ levels in major urban and industrial areas, VOC emission controls should be added to the current $NOx$-$SO_2$-PM policy". Does your research have similar insights, or can you make other recommendations that could help policymakers?*

**Response:**

Thanks for reviewer's suggestion. Our suggestion is consistent with Liu and Wang (2020), we hope that the government should not focus on the control of $PM_{2.5}$ pollution ($NOx$-$SO_2$-PM policy),

but should pay attention to the synergistic control of multiple pollutants such as $O_3$ and $PM_{2.5}$.

**Technical corrections:**

*1. L211, "353 stations" - > "353 meteorological stations"*
   **Response:**
       Thanks for your suggestion. We have added the "meteorological" in the revised manuscript.
   (**Page 9, Line 235**)

*2. Figure S5, "from 2013to" - > "from 2013 to"*
   **Response:**
       Thanks for your suggestion. We have changed the expression in the revised manuscript.
   (**Page 11 in supporting information**)

*3. Figure 6, "on the right side of each panel" - > "on the upper right side of each panel"*
   **Response:**
       According to the reviewer's suggestion, we have changed the expression in the revised manuscript. (**Page 37, Line 859**)

*4. Data and code availability should be added.*
   **Response:**
       According to the reviewer's suggestion, we have added the "Data availability" section in the revised manuscript. (**Page 22, Line 593-600**)

*Reference:*

*Gao, J., Li, Y., Xie, Z., Hu, B., Wang, L., Bao, F., and Fan, S.: The impact of the aerosol reduction on the worsening ozone pollution over the Beijing-Tianjin-Hebei region via influencing photolysis rates, Sci. Total Environ., 821, 153197, https://doi.org/10.1016/j.scitotenv.2022.153197, 2022.*

*Liu, Y. and Wang, T.: Worsening urban ozone pollution in China from 2013 to 2017 – Part 2: The effects of emission changes and implications for multi-pollutant control, Atmospheric Chem. Phys., 20, 6323–6337, https://doi.org/10.5194/acp-20-6323-2020, 2020.*

*Lou, S., Liao, H., and Zhu, B.: Impacts of aerosols on surface-layer ozone concentrations in China through heterogeneous reactions and changes in photolysis rates, Atmos. Environ., 85, 123–138, https://doi.org/10.1016/j.atmosenv.2013.12.004, 2014.*

**Reference:**

Atkinson, R.: Atmospheric chemistry of VOCs and NOx, Atmos Environ., 34, 2063–2101, https://doi.org/10.1016/S1352-2310(99)00460-4, 2000.

Edson, C. T., Ivan, H.-P. and Alberto, M.: Use of combined observational- and model-derived photochemical indicators to assess the $O_3$-$NO_x$-VOC System sensitivity in urban areas, Atmosphere., 8, 22.

https://doi.org/10.3390/ atmos8020022, 2017.

Gipson, G. L.: Science algorithms of the EPA Models-3 community multiscale air quality (CMAQ) modeling system: Chapter 16, process analysis, edited by: Byun, D. W. and Ching, J. K. S., Reported No. EPA/600/R-99/030, U.S. Environmental Protection Agency, Office of Research and Development, Washington, D.C., 1999.

Hong, C., Zhang, Q., Zhang, Y., Davis, S. J., Zhang, X., Tong, D., Guan, D., Liu, Z., and He, K.: Weakening aerosol direct radiative effects mitigate climate penalty on Chinese air quality, Nat. Clim. Change, 10, 845–850, https://doi.org/10.1038/s41558-020-0840-y, 2020.

Lakey, P. S. J., George, I. J., Whalley, L. K., Baeza-Romero, M. T., and Heard, D. E.: Measurements of the $HO_2$ Uptake Coefficients onto Single Component Organic Aerosols, Environmental Science & Technology, 49, 4878-4885, 10.1021/acs.est.5b00948, 2015.

Li, K., Chen, L., Ying, F., White, S. J., Jang, C., Wu, X., Gao, X., Hong, S., Shen, J., Azzi, M. and Cen, K: Meteorological and chemical impacts on ozone formation: a case study in Hangzhou, China, Atmos. Res., 196, https://doi.org/10.1016/ j.atmosres.2017.06.003, 2017.

Li, K., Jacob, D. J., Liao, H., Shen, L., Zhang, Q., and Bates, K. H.: Anthropogenic drivers of 2013–2017 trends in summer surface ozone in China, P. Natl. Acad. Sci. USA, 116, 422–427, https://doi.org/10.1073/pnas.1812168116, 2019.

Liu, Y. and Wang, T.: Worsening urban ozone pollution in China from 2013 to 2017 – Part 1: The complex and varying roles of meteorology, Atmos. Chem. Phys., 20, 6305–6321, https://doi.org/10.5194/acp-20-6305-2020, 2020a.

Liu, Y. and Wang, T.: Worsening urban ozone pollution in China from 2013 to 2017 – Part 2: The effects of emission changes and implications for multi-pollutant control, Atmos. Chem. Phys., 20, 6323–6337, https://doi.org/10.5194/acp-20-6323-2020, 2020b.

Lo, J. C.-F., Yang, Z. L., and Pielke Sr, R. A.: Assessment of three dynamical climate downscaling methods using the Weather Research and Forecasting (WRF) model, J. Geophys. Res., 113, D09112, https://doi.org/10.1029/2007jd009216, 2008.

Qiu, Y., Liao, H., Zhang, R., and Hu, J.: Simulated impacts of direct radiative effects of scattering and absorbing aerosols on surface layer aerosol concentrations in China during a heavily polluted event in February 2014, J. Geophys. Res. Atmos., 122, 5955–5975, doi:10.1002/2016JD026309, 2017.

Seinfeld, J. H. and Pandis, S. N.: Atmospheric Chemistry and Physics: from Air Pollution to Climate Change, second ed., John Wiley and Sons, 2006.

Shao, M., Wang, W. J., Yuan, B., Parrish, D. D., Li, X., Lu, K. D., Wu, L. L., Wang, X. M., Mo, Z. W., Yang, S. X., Peng, Y. W., Kuang, Y., Chen, W. H., Hu, M., Zeng, L. M., Su, H., Cheng, Y. F., Zheng, J. Y., Zhang, Y. H.: Quantifying the role of $PM_{2.5}$ dropping in variations of ground-level ozone: Inter-comparison between Beijing and Los Angeles, Sci. Total Environ., https://doi.org/10.1016/j.scitotenv.2021.147712, 2021.

Shu, L., Wang, T., Han, H., Xie, M., Chen, P., Li, M., and Wu, H.: Summertime ozone pollution in the Yangtze River Delta of eastern China during 2013–2017: Synoptic impacts and source apportionment, Environ. Pollut., 257, 113631, https://doi.org/10.1016/j.envpol.2019.113631, 2020.

Shu, L., Xie, M., Wang, T., Gao, D., Chen, P., Han, Y., Li, S., Zhuang, B., and Li, M.: Integrated studies of a regional ozone pollution synthetically affected by subtropical high and typhoon system in the Yangtze River Delta region, China, Atmos. Chem. Phys., 16, 15801–15819, https://doi.org/10.5194/acp-16-15801- 2016, 2016.

Sillman, S.: The relation between ozone, $NO_x$ and hydrocarbons in urban and polluted rural environments, Atmos. Environ., 33, 1821-1845, https://doi.org/ 10.1016/S1352-2310(98)00345-8, 1999.

Taketani, F., Kanaya, Y., and Akimoto, H.: Heterogeneous loss of $HO_2$ by KCl, synthetic sea salt, and natural seawater aerosol particles, Atmospheric Environment, 43, 1660-1665, 2009.

Tan Z, Hofzumahaus A, Lu K, Brown SS, Holland F, Huey LG, et al. No Evidence for a Significant Impact of Heterogeneous Chemistry on Radical Concentrations in the North China Plain in Summer 2014. Environ. Sci. Technolc. 54, 5973-5979, 2020.

Wang, N., Lyu, X., Deng, X., Huang, X., Jiang, F., and Ding, A.: Aggravating O3 pollution due to NOx emission control in eastern China, Sci. Total Environ., 677, 732–744, 2019.

Yang, H., Chen, L., Liao, H., Zhu, J., Wang, W., and Li, X.: Impacts of aerosol–photolysis interaction and aerosol–radiation feedback on surface-layer ozone in North China during multi-pollutant air pollution episodes, Atmos. Chem. Phys., 22, 4101–4116, https://doi.org/10.5194/acp-22-4101-2022, 2022.

Zhang, B., Wang, Y., and Hao, J.: Simulating aerosol–radiation–cloud feedbacks on meteorology and air quality over eastern China under severe haze conditionsin winter, Atmos. Chem. Phys., 15, 2387–2404, https://doi.org/10.5194/acp-15-2387-2015, 2015.

Zhao, B., Wang, S., Donahue, N. M., Chuang, W., Ruiz, L. H., Ng, N. L., Wang, Y., and Hao, J.: Evaluation of One-Dimensional and Two-Dimensional Volatility Basis Sets in Simulating the Aging of Secondary Organic Aerosol with Smog-Chamber Experiments, Environ. Sci. Technol., 49, 2245–2254, doi:10.1021/es5048914, 2015.

Zhu, J., Chen, L., Liao, H., Yang, H., Yang, Y., and Yue, X.: Enhanced PM2.5 Decreases and O3 Increases in China During COVID-19 Lockdown by Aerosol-Radiation Feedback, Geophys. Res. Lett., 48, https://doi.org/10.1029/2020GL090260, 2021.

Zhou, M., Zhang, L., Chen, D., Gu, Y., Fu, T.-M., Gao, M., Zhao, Y., Lu, X. and Zhao, B.: The impact of aerosol-radiation interactions on the effectiveness of emission control measures, Environmental Research Letters, 14(2), 024002, https://doi.org/10.1088/1748-9326/aaf27d, 2019.

Zou Q, Song H, Tang M, Lu K. Measurements of $HO_2$ uptake coefficient on aqueous $(NH_4)_2SO_4$ aerosol using aerosol flow tube with LIF system. Chinese Chemical Letters 2019; 30: 2236-2240.

**Thank you very much for your comments and suggestions.**

---

## Author Comment (AC2)

**Response to Comments of Reviewer #3**

**(comments in *italics*)**

**Manuscript number:** EGUSPHERE-2023-2393

**Title:** Weakened aerosol-radiation interaction exacerbating ozone pollution in eastern China since China's clean air actions

*The manuscript focuses on the aerosol-radiation interaction (ARI), discussing how this process has changed in the context of the abrupt aerosol decrease in East China during 2013-2017, and evaluates its contribution to the recent ozone increase in China. ARI is divided into aerosol-photolysis interaction (API) and aerosol-radiation feedback (ARF), with the WRF-Chem model used to quantify these impacts. The authors have found non-negligible ozone increase resulting from the aerosol decrease through the API and ARF processes, which has implications for the synergistic control of aerosol and ozone. This is an interesting topic and I believe it can make a novel contribution to the community. However, several important aspects need to be addressed before it can be published in ACP.*

**Response:**

Thanks to the reviewer for the valuable comments and suggestions which are very helpful for us to improve our manuscript. We have revised the manuscript carefully, as described in our point-to-point responses to the comments.

**General comments:**

1. *The study focuses on aerosol-radiation interaction (ARI), which is split into two parts: the direct aerosol impact on radiation through scattering and absorbing (API) and the subsequent feedback on meteorology (ARF), with both influencing ozone concentrations. However, the Introduction Section could do a better job at breaking down these concepts. A detailed explanation of the distinctions between API and ARF would aid comprehension. Also, elucidating the specific ARF-related meteorological variables and their influences on ozone concentrations would be beneficial. Regarding the cited papers, such as Hong et al. (2020) and Zhu et al. (2021), the authors may consider including additional information about which ARF-related meteorological factors have been identified as important in affecting ozone concentrations.*

**Response:**

Thanks to the reviewer for the valuable comments and suggestions, we have added this information in the revised manuscript as follows: "API can affect $O_3$ directly by reducing the photochemical reactions, which weaken the chemical contribution and reduce the surface $O_3$ concentrations. ARF indirectly affects $O_3$ concentrations by altering meteorological variables, e.g. by reducing the height of the planetary boundary layer. The suppressed planetary boundary layer can weaken the vertical mixing of $O_3$ by turbulence and affect the concentration of $O_3$ precursors. Hong et al. (2020) used WRF-CMAQ in conjunction with

future emission scenarios to find that weakened ARF due to reduced aerosol concentration has either negative or positive impacts on the daily maximum 1-h average $O_3$ concentration in eastern China from 2010 to 2050 due to the changed precursor level caused by the weakened ARF. By using WRF-CMAQ, Liu and Wang (2020b) reported that weakened API could increase the MDA8 $O_3$ concentrations by 0.3 ppb in urban areas from 2013 to 2017. Zhu et al. (2021) used WRF-Chem to investigate the impact of weakened ARF on air pollutants over NCP during COVID-19 lockdown and reported that the weakened ARF would increase the $O_3$ concentrations by 7.8% due to the increased northwesterly and planetary boundary layer height caused by the weakened ARF." (**Page 4-5, Line 95-110**)

2. *In Section 3.2, could the authors talk more about how well the model is doing in reproducing the observed decrease in $PM_{2.5}$ levels from 2013-2017. This analysis is crucial for assessing whether the model's effectively capturing the weakening of ARI.*

**Response:**

Thanks for your suggestion. Figure R1 demonstrates the spatial distribution of changed summer (left) and winter (right) surface (a, b) $PM_{2.5}$ and (c, d) MDA8 $O_3$ from 2013 to 2017. As shown in Figs. R1(a) and R1(b), the observed concentrations of $PM_{2.5}$ in eastern China are significantly reduced both in summer (-16.2 µg m$^{-3}$) and winter (-56.0 µg m$^{-3}$), and these changes can be well captured by the model (-14.3 µg m$^{-3}$ for summer and -49.8 µg m$^{-3}$ for winter). Therefore, the model can reproduce the observed decrease in $PM_{2.5}$ levels from 2013 to 2017. As shown in Figs. R1(c) and R1(d), the model reasonably well reproduces the seasonal patterns of changed surface MDA8 $O_3$ over the eastern China during summer and winter from 2013 to 2017. In summer, both the observations and simulations show the increased (decreased) MDA8 $O_3$ in YRD (PRD and SCB), while the model can not simulate the positive changes in MDA8 $O_3$ over BTH, and the potential reasons may be that this study did not consider the effect of changes in aerosol heterogeneous reactions. Li et al. (2019) found that the weakened uptake of $HO_2$ on aerosol surfaces was the main reason for the $O_3$ increase over BTH. In contrast to the changes in summer, observed MDA8 $O_3$ in winter generally increased over the eastern China, which can be well reproduced by the model. (**Page 12, Line 308-324**)

According to the reviewer's comments, Figure R1 is added in the model evaluation section. (**Figure 3**)

[Figure]

**Figure R1.** Spatial distribution of changed summer (left) and winter (right) surface (a, b) PM$_{2.5}$ and (c, d) MDA8 O$_3$ from 2013 to 2017.

3. *Section 4 needs to be better organized for clarity. I've outlined some areas for consideration:*

3.1. *The titles suggest Section 4.1 should focus on ΔO$_3$\_MET and ΔO$_3$\_EMI, while 4.2 should be devoted to ΔO$_3$\_ΔARI\_EMI. However, there is content overlap since 4.1 also examines ΔO$_3$\_ΔARI\_EMI, which obscures the distinctions between the two subsections.*

**Response:**

Thanks for your suggestion. We have changed this in revised manuscript. Section 4.1 focuses only on the ΔO$_3$\_MET and ΔO$_3$\_EMI, and the results of the ΔO$_3$\_ΔARI\_EMI in urban areas have been moved to Section 4.2. (**Page 12-13, Line 326-349**)

3.2. *Section 4.1 discusses ΔO$_3$\_MET, ΔO$_3$\_EMI, and ΔO$_3$\_ΔARI\_EMI at sparse polluted grids (so-called urban areas) while 4.2 talks about ΔO$_3$\_ΔARI\_EMI in term of regional averages. It is unclear why the discussion about ΔO$_3$\_MET and ΔO$_3$\_EMI focuses only on urban polluted regions. Also, the rationale for addressing urban ΔO$_3$\_ΔARI\_EMI prior to regional averages is not evident, particularly when urban results mirror the regional ones, though more pronounced. I recommend relocating the OBS-SIM ozone change comparison from Section 4.1 to Section 3.2 (to combine it with PM$_{2.5}$ change evaluation) and discussing regional ΔO$_3$\_ΔARI\_EMI before the urban analysis.*

**Response:**

Thanks for your suggestion. The comparison of O$_3$ change from 2013 to 2017 has been combined with the comparison of PM$_{2.5}$ change in Section 3. The detailed information can be found in the answer to your second question.

According to review's suggestion, in the revised manuscript we first discussed the effects of weakened ARI on O$_3$ at the regional level, and then in urban areas. (**Page 14-18, Line 383-495**)

*3.3.* *Section 4.3 and Figure 7 are quite similar to Section 4.2 and Figure 5. Please consider merging Sections 4.2 and 4.3.*

**Response:**

Thanks for your suggestion. We've combined these two sections in the revised manuscript.

*4.* *Could the authors explain why $\Delta O_3\_\Delta ARI\_EMI$ displays a much steeper spatial gradient in summer compared to winter (Fig. 5), whereas the $PM_{2.5}$ change suggest the opposite pattern (Fig. S8)? How does meteorology contribute to this discrepancy? Moreover, why does summertime $\Delta O_3\_\Delta ARI\_EMI$ exhibit both positive (e.g., NCP) and negative (e.g., Shandong province) values, even though the $PM_{2.5}$ decreases universally?*

**Response:**

The reason may be that the solar radiation flux reaches its maximum in summer seasons. The changes in meteorological variables are larger in summer than in winter due to the weakened ARI, despite the substantial decrease in aerosol concentrations during winter. Meteorology is likely to be a major contributor to this discrepancy.

Although the concentration of $PM_{2.5}$ is reduced uniformly, the changes in the components of $PM_{2.5}$ are different in different locations, resulting in different changes in single scattering albedo (SSA). As shown in Fig. R2, SSA did not change in NCP, but became smaller in Shandong Province, which may be the reason for the different changes in $O_3$ in these two regions. Furthermore, Fig. S7(b3) and S7(c3) show that weakened aerosol-radiation interaction leads to a decrease in $T_2$ but an increase in $RH_2$ over Shandong, which is also unfavourable for $O_3$ production. This could also be one of the reasons why weakened aerosol-radiation interaction leads to $O_3$ reduction in Shandong Province.

[Figure]

**Figure R2.** Spatial distribution of **(a, d)** scattering aerosol, **(b, e)** absorbing aerosol, and **(c, f)** single scattering albedo (SSA) of BASE_17E17M (upper) and BASE_13E17M (bottom) cases.

5. *From my understanding, the reduced impact of ARI on ozone is a component of the anthropogenic impact on ozone, since the reduction in ARI results from changes in anthropogenic emissions. However, the phrasing in Lines 396-398 and abstract (specifically the use of "superimposed") suggest that $\Delta O_3\_\Delta ARI\_EMI$ is and additional, separate effect rather than being nested within the broader anthropogenic impact on ozone. Please clarify.*

**Response:**

Thanks for your suggestion. Figure R3 shows the changed summer and winter surface-layer MDA8 $O_3$ concentrations caused by anthropogenic emission reduction from 2013 to 2017 with ($\Delta O_3\_EMI$) and without ($\Delta O_3\_NOARI$) ARI, including the effects of weakened ARI on the effectiveness of emission reduction for $O_3$ air quality ($\Delta O_3\_\Delta ARI\_EMI$, which is also equal to $\Delta O_3\_EMI$ minus $\Delta O_3\_NOARI$). As shown in Figs. R3(a1) and R3(a4), the surface-layer MDA8 $O_3$ concentrations increased in urban areas during summer and increased uniformly in winter due to anthropogenic emission reduction from 2013 to 2017 without the impact of ARI. The plots in the second column (Figs. R3(a2) and R3(a5)) are the same as R3(a1) and R3(a4) except that the impact of ARI is applied. When the effect of ARI is considered, the concentrations of MDA8 $O_3$ are increased more than that when ARI is not considered. The differences between plots in second column and first column are the consequences of weakened ARI resulted from anthropogenic emission reduction on MDA8 $O_3$ concentrations. As shown in Figs. R3(a3) and R3(a6), the concentrations of MDA8 $O_3$ are increased in both summer and winter over eastern China. Therefore, $\Delta O_3\_\Delta ARI\_EMI$ makes the superimposed impact on the effectiveness of anthropogenic emission reduction for the increased MDA8 $O_3$ concentrations from 2013 to 2017 over eastern China.

[Figure]

**Figure R3.** Spatial distribution of changed summer (upper) and winter (bottom) surface-layer MDA8 $O_3$ concentrations from sensitivity simulations. **(a1, a4)** Effects of anthropogenic emission reduction on MDA8 $O_3$ without ARI. **(a2, a5)** Effects of anthropogenic emission reduction on MDA8 $O_3$ with ARI. **(a3, a6)** Effects of weakened ARI on the effectiveness of emission reduction for $O_3$ air quality.

6. *In the Abstract, needs to explicitly clarify that the numbers presented are derived from different analysis. Lines 28-29 are for sparse polluted grids, while Lines 33-35 are for regional averages. Otherwise, readers may erroneously interpret the ratio between the numbers in Lines 33-35 and Lines 28-29 as the contribution of ARI to the total anthropogenic impacts.*

**Response:**

   Thanks for your suggestion. We've added this information in the revised manuscript as follows:

   "Sensitivity experiments show that the decreased anthropogenic emissions play a more prominent role for the increased MDA8 $O_3$ both in summer (+1.96 ppb vs. +0.07 ppb) and winter (+3.56 ppb vs. -1.08 ppb) than the impacts of changed meteorological conditions in urban areas. (**Page 2, Line 27-31**)

   The weakened ARI due to decreased anthropogenic emission aggravates the summer (winter) $O_3$ pollution by +0.81 ppb (+0.63 ppb) averaged over eastern China, with weakened API and ARF contributing 55.6% (61.9%) and 44.4% (38.1%), respectively. This superimposed effect is more significant for urban areas during summer (+1.77 ppb). (**Page 2, Line 33-37**)"

**Specific comments:**

1. Line 61, natural emissions are also an important precursor source. Please clarify.

   **Response:**

   According to the reviewer's suggestion, we have changed the expression in the revised manuscript. (**Page 3, Line 65-67**)

2. Section 3.2, it should be "Fig. 2" instead of "Figs. 2". Similar typos are found in other places, e.g., Line 290, 302, 348. Please check.

   **Response:**

   Thanks for your suggestion. Since it's followed by a plural, we use "Figs".

3. Line 293, delete "will".

   **Response:**

   Deleted.

4. Lines 310-312 and figure 4, please clarify in the figure caption that ARI_EMI can be obtained by summing the bars of API_EMI and ARF_EMI.

   **Response:**

   Thanks for your suggestion. We have defined the $\Delta O3\_\Delta ARI\_EMI = \Delta O3\_\Delta ARF\_EMI + \Delta O3\_\Delta API\_EMI$ in the revised manuscript. (**Page 15, Line 405-406**)

5. Lines 353-354 and figure 5, the numbers mentioned in the text are inconsistent with those presented in the figure. Please correct.

   **Response:**

   Correct.

6. Figure 6, the first x-axis label should be "ARI" instead of "ALL".

   **Response:**

   Thanks for your suggestion. We have changed the expression in the revised manuscript. (**Page 37**)

**Reference:**

Hong, C., Zhang, Q., Zhang, Y., Davis, S. J., Zhang, X., Tong, D., Guan, D., Liu, Z., and He, K.: Weakening aerosol direct radiative effects mitigate climate penalty on Chinese air quality, Nat. Clim. Change, 10, 845–850, https://doi.org/10.1038/s41558-020-0840-y, 2020.

Li, K., Jacob, D. J., Liao, H., Shen, L., Zhang, Q., and Bates, K. H.: Anthropogenic Drivers of 2013–2017 Trends in Summer Surface Ozone in China, P. Natl. Acad. Sci. USA, 116, 422–427, https://doi.org/10.1073/pnas.1812168116, 2019.

Liu, Y. and Wang, T.: Worsening urban ozone pollution in China from 2013 to 2017 – Part 2: The effects of emission changes and implications for multi-pollutant control, Atmos. Chem. Phys., 20, 6323–6337, https://doi.org/10.5194/acp-20-6323-2020, 2020b.

Zhu, J., Chen, L., Liao, H., Yang, H., Yang, Y., and Yue, X.: Enhanced PM2.5 Decreases and O3 Increases in China During COVID-19 Lockdown by Aerosol-Radiation Feedback, Geophys. Res. Lett., 48, https://doi.org/10.1029/2020GL090260, 2021.

**Thank you very much for your comments and suggestions.**

---

## Author Comment (AC3)

**Response to Comments of Reviewer #2**

**(comments in *italics*)**

**Manuscript number:** EGUSPHERE-2023-2393

**Title:** Weakened aerosol-radiation interaction exacerbating ozone pollution in eastern China since China's clean air actions

*This study examines the role of aerosol-radiation interaction (ARI), decomposed into aerosol-photolysis interaction (API) and aerosol-radiation feedback (ARF) on surface ozone concentration in China. Surface ozone increased remarkable in eastern China, contrasting the dramatic decline of $PM_{2.5}$ concentrations. It is therefore necessary to investigate the reasons for the ozone increase. The study found that reduced ARI due to decreased PM concentrations contributes to ozone production, with API playing a more important role than ARF. The regional differences are also briefly discussed. I think this is a nice study that is helpful in understanding the recent ozone increase in China. I only have a few minor comments.*

**Response:**

Thanks to the reviewer for the valuable comments and suggestions which are very helpful for us to improve our manuscript. We have revised the manuscript carefully, as described in our point-to-point responses to the comments.

1. *A previous study seemed to indicate that chemical processes associated with $PM_{2.5}$ reduction, i.e., reduced removing rate of hydroperoxy radicals, is the main reason for the ozone increase in eastern China (Li et al., 2019, PNAS). I wonder how this effect compare to the ARI discussed in this study?*

**Response:**

As Li et al. (2019) did not directly quantify the extent of $O_3$ increase by weakened aerosol heterogeneous reactions, we use the results of Liu and Wang. (2020) for comparison. The increased MDA8 $O_3$ concentration over urban areas in summer caused by weakened aerosol-radiation interaction in this study is 1.77 ppb, which is compared to the value of 2.12 ppb increase caused by weakened aerosol heterogeneous reactions quantified by Liu and Wang (2020). According to the reviewer's comments, we have added this sentence in the revised manuscript. (**Page 18, Line 485-488**)

2. *In the WRF-Chem experiments, the authors zeroed off aerosol optical properties to exclude ARF. I wonder if aerosol microphysical properties are still included? This may affect cloud properties and still impact the radiation budget.*

**Response:**

The effects of aerosols on microphysical properties were not consider in this work. The most common approach to assessing the impact of aerosol-cloud interactions on air quality in model simulation is to assume a prescribed vertically uniform cloud droplet number concentration (Zhang et al., 2015; Zhao et al., 2017). In this study, we turned off aerosol optical properties in the

optical module which could not affect the cloud properties.

Figure R1 shows the spatial distributions of simulated summer and winter cloud droplet number concentration (CDNC) from BASE_17E17M and NOALL_17E17M cases in the daytime (08:00–17:00 LST). Analyzing Fig. R1, the CDNC distribution and concentration of BASE and NOALL has barely changed. Therefore, we zeroed off aerosol optical properties to exclude ARI with less impact on the cloud.

[Figure]

**Figure R1.** Spatial distributions of simulated summer (upper) and winter (bottom) cloud droplet number concentration (CDNC) from BASE_17E17M and NOALL_17E17M cases in the daytime (08:00–17:00 LST).

3. *Section 3.2, model evaluation: why not also evaluate VOCs, which is also an important precursor for ozone?*

**Response:**

Thanks for reviewer's suggestion. In this study, we did not evaluate VOCs due to the lack of measurements of VOCs over the China. However, the China's Ministry of Environmental Protection will include VOCs as a routine monitoring object in the future. Therefore, we will include this comparison in our future work.

4. *Line 87 and associated discussions: Does ARI always suppress $O_3$ formation? Could the change the meteorological variables through ARF increase $O_3$ concentration, say by reducing RH or increasing regional transport?*

**Response:**

Yang et al. (2022) reported that ARF reduced the planetary boundary layer height in North China, leading to an increase in VOCs and $NO_x$ concentrations, which is favorable for ozone chemical production. Gao et al. (2018) also found that ARF can enhance ozone

chemical production through this pathway. Therefore, ARF can increase O$_3$ concentration by influencing the meteorological variables, e.g. by reducing the height of the planetary boundary layer.

5. *I suggest the authors discuss more about the summer-winter differences. Wintertime has much less radiation and lower temperature, so ARI is in general much lower. In summer, meteorology seems to make large contributions than emission changes (Figure 4, left column), what might be the reason?*

**Response:**

Focusing on the four developed city clusters, compared with 2013, the meteorological conditions in the summer of 2017 promoted the generation of O$_3$ in the YRD region (Fig. R2(a3)), but suppressed the generation of O$_3$ in the BTH (Fig. R2(a2)), PRD (Fig. R2(a4)) and SCB (Fig. R2(a5)) regions. In PRD and SCB, the changes in MDA8 O$_3$ due to meteorology even have a greater impact than that by emission changes, which highlights the significant role of meteorology on summer O$_3$ variations. **(Page 13, Line 343-349)**

According to the comments of Reviewer#1, another three widely used chemical mechanisms, i.e., RADM2 gas-phase chemistry coupled with MADE/SORGAM aerosol module (RADM2-MADE/SORGAM for short), CBMZ gas-phase chemistry coupled with MADE/SORGAM aerosol module (CBMZ-MADE/SORGAM for short), and MOZART gas-phase chemistry coupled with MOSAIC aerosol module (MOZART-MOSAIC for short), that include SOA formation are also applied to assess the impact of aerosol-radiation interaction (ARI) on O$_3$ during summer and winter is added in the discussion section. (**Page 18-19, Line 497-536**)

In summer, solar radiation flux reaches its maximum and atmospheric temperature are also higher than that in winter. The atmospheric warming can alter tropospheric O$_3$ concentrations by modulating the chemical kinetic, dynamic processes or biogenic emissions. Warmer temperatures often coincide with other meteorological conditions favorable to O$_3$ production, such as stagnation air and reduced cloud cover (Vukovich, 1995). This may be the reason why meteorological effect on O$_3$ is greater than that by emissions changes.

[Figure]

**Figure R2.** The observed (OBS, black bars) and simulated (SIM, red bars) changes in (left) summer and (right) winter surface-layer MDA8 $O_3$ from 2013 to 2017. Contributions of changed meteorological conditions alone (MET, blue bars), changed anthropogenic emissions alone (EMI, purple bars), changed aerosol-photolysis interaction alone ($\Delta$API_EMI, green bars), and changed aerosol-radiation feedback alone ($\Delta$ARF_EMI, cyan bars) are also shown. Observations are calculated from the monitoring sites in the analyzed region, while the corresponding gridded simulations are averaged for SIM. **(a1-b1)**, **(a2-b2)**, **(a3-b3)**, **(a4-b4)** and **(a5-b5)** represent the urban areas in eastern China, Beijing-Tianjin-Hebei (BTH), Yangtze River Delta (YRD), Pearl River Delta (PRD), and Sichuan Basin (SCB), respectively.

6. *Figure 4: model seems to significantly underestimate the ozone change in BTH for summer (Figure 4a2). This area experienced the most ozone increases in the past decade. So it is important for the model to correctly represent ozone trend in this region. What might be the reason for this significant bias?*

**Response:**

Thanks for your suggestion. The reason for the underestimation over BTH in summer may be that this study did not consider the effect of changes in aerosol heterogeneous reactions, due to the uncertainty of the heterogeneous uptake value used in the numerical simulation. Li et al. (2019) found that the weakened uptake of $HO_2$ on aerosol surfaces was

the main reason for the $O_3$ increase over BTH. Therefore, the contributions of aerosol heterogeneous reactions to $O_3$ air quality will be discussed detailedly in our future work.

7.  *Finally, the effects of API and ARF may not be independent, i.e., there may be nonlinear interaction between the two effects. This should be noted and discussed.*

**Response:**

Thanks for the reviewer's suggestion. A discussion of the separate treatment of API and ARF in this study has been added in the revised manuscript as follows: "There may be an interaction between API and ARF. However, in this study we discuss the role of API and ARF separately, which may ignore the effects of interactions between API and ARF on $O_3$. This may affect our results, and we will discuss their interaction in our future studies." (**Page 20, Line 557-560**)

**Reference:**

Gao, J. H., Zhu, B., Xiao, H., Kang, H. Q., Pan, C., Wang, D. D., and Wang, H. L.: Effects of black carbon and boundary layer interaction on surface ozone in Nanjing, China, Atmos. Chem. Phys., 18, 7081–7094, https://doi.org/10.5194/acp-18-7081-2018, 2018.

Li, K., Jacob, D. J., Liao, H., Shen, L., Zhang, Q., and Bates, K. H.: Anthropogenic drivers of 2013–2017 trends in summer surface ozone in China, P. Natl. Acad. Sci. USA, 116, 422–427, https://doi.org/10.1073/pnas.1812168116, 2019.

Liu, Y. and Wang, T.: Worsening urban ozone pollution in China from 2013 to 2017 – Part 2: The effects of emission changes and implications for multi-pollutant control, Atmos. Chem. Phys., 20, 6323–6337, https://doi.org/10.5194/acp-20-6323-2020, 2020.

Vukovich F. M.: Regional-scale boundary layer ozone variations in the eastern United States and their association with meteorological variations, Atmos. Environ., 29, 2259-2273, 1995.

Yang, H., Chen, L., Liao, H., Zhu, J., Wang, W., and Li, X.: Impacts of aerosol–photolysis interaction and aerosol–radiation feedback on surface-layer ozone in North China during multi-pollutant air pollution episodes, Atmos. Chem. Phys., 22, 4101–4116, https://doi.org/10.5194/acp-22-4101-2022, 2022.

Zhang, B., Wang, Y., and Hao, J.: Simulating aerosol–radiation–cloud feedbacks on meteorology and air quality over eastern China under severe haze conditionsin winter, Atmos. Chem. Phys., 15, 2387–2404, https://doi.org/10.5194/acp-15-2387-2015, 2015.

Zhao, B., Liou, K.-N., Gu, Y., Li, Q., Jiang, J. H., Su, H., He, C., Tseng, H.-L. R., Wang, S., Liu, R., Qi, L., Lee, W.-L., and Hao, J.: Enhanced $PM_{2.5}$ pollution in China due to aerosol-cloud interactions, Scient. Rep., 7, 4453, https://doi.org/10.1038/s41598-017-04096-8, 2017.

**Thank you very much for your comments and suggestions.**